



# Seasonal partitioning of precipitation between streamflow and evapotranspiration, inferred from end-member splitting analysis

James W. Kirchner[1,2,3] and Scott T. Allen[1,4]

[1]Dept. of Environmental Systems Science, ETH Zurich, 8092 Zurich, Switzerland
[2]Swiss Federal Research Institute WSL, 8903 Birmensdorf, Switzerland
[3]Dept. of Earth and Planetary Science, University of California, Berkeley, CA 94720, USA
[4]Dept. of Geology and Geophysics, University of Utah, Salt Lake City, UT, 84112, USA

*Correspondence to*: James W. Kirchner (kirchner@ethz.ch)

**Abstract.** A key attribute of the terrestrial water cycle is the partitioning of precipitation into its two ultimate fates: "green
water" that is evaporated or transpired back to the atmosphere, and "blue water" that is discharged to stream channels.
Measuring this partitioning is difficult, particularly on seasonal timescales. End-member mixing analysis has been widely
used to quantify streamflow as a mixture of isotopically distinct sources, but knowing where streamwater comes from is not
the same as knowing where precipitation goes, and this latter question is the one we seek to answer. Here we introduce
"end-member splitting analysis", which uses isotopic tracers and water flux measurements to quantify how isotopically
distinct inputs (such as summer vs. winter precipitation) are partitioned into different ultimate outputs (such as
evapotranspiration and summer vs. winter streamflow). End-member splitting analysis has modest data requirements, and
can potentially be applied in many different catchment settings. We illustrate this data-driven, model-independent approach
with publicly available biweekly isotope time series from Hubbard Brook Watershed 3. A marked seasonal shift in isotopic
composition allows us to distinguish rainy-season (April-November) and snowy-season (December-March) precipitation,
and to trace their respective fates. End-member splitting shows that about one-sixth (18±2%) of rainy-season precipitation is
discharged during the snowy season, but this accounts for over half (60±9%) of snowy-season streamflow. By contrast,
most (55±13%) snowy-season precipitation becomes streamflow during the rainy season, where it accounts for 38±9% of
rainy-season streamflow. Our analysis thus shows that significant fractions of each season's streamflow originated as the
other season's precipitation, implying significant inter-seasonal water storage within the catchment, as both groundwater and
snowpack. End-member splitting can also quantify how much of each season's precipitation is eventually evapotranspired.
At Watershed 3, we find that only about half (44±8%) of rainy-season precipitation evapotranspires, but almost all (85±15%)
evapotranspiration originates as rainy-season precipitation, implying that there is relatively little inter-seasonal water storage
supplying evapotranspiration. We show how results from this new technique can be combined with young water fractions
(calculated from seasonal isotope cycles in precipitation and streamflow) and new water fractions (calculated from
correlations between precipitation and streamflow isotope fluctuations) to infer how precipitation is partitioned on multiple
time scales. This proof-of-concept study demonstrates that end-member mixing and splitting yield different, but





complementary, insights into catchment-scale partitioning of precipitation into blue water and green water. It could thus help in gauging the vulnerability of both water resources and terrestrial ecosystems to changes in seasonal precipitation.

**1 Introduction: end-member mixing and end-member splitting**

End-member mixing analysis has been widely used in isotope hydrograph separation, as well as in other applications that seek to interpret environmental flows as mixtures of chemically or isotopically distinct end-member sources (see Klaus and McDonnell, 2013, and references therein). The simplest form of end-member mixing analysis uses a single conservative tracer to estimate the fractions of two sources in a mixture (see Fig. 1). It is derived from the mass balances for the water
and tracer,

$$q_{A \to M} + q_{B \to M} = Q_M \qquad (1)$$

and

$$q_{A \to M}\, \bar{\delta}_A + q_{B \to M}\, \bar{\delta}_B = Q_M\, \bar{\delta}_M \quad , \qquad (2)$$

where $q_{A \to M}$ and $q_{B \to M}$ denote fluxes from end-members A and B to a mixture M whose total flux is $Q_M$, and the volume-
weighted isotope signatures (or tracer solute concentrations) in these three fluxes are $\bar{\delta}_A$, $\bar{\delta}_B$, and $\bar{\delta}_M$, respectively. These equations embody the two essential assumptions of end-member mixing analysis: that the mixture M is sourced from (and only from) A and B (Eq. 1), and that the tracer is conservative, with no other sources or sinks that alter the tracer signatures $\bar{\delta}_A$ and $\bar{\delta}_B$ between the end-members A and B and the mixture M (Eq. 2). Simultaneously solving Eqs. (1) and (2) yields the well-known end-member mixing equations,

$$f_{M \leftarrow A} = \frac{q_{A \to M}}{Q_M} = \frac{\bar{\delta}_M - \bar{\delta}_B}{\bar{\delta}_A - \bar{\delta}_B} \quad \text{and} \quad f_{M \leftarrow B} = 1 - f_A = \frac{q_{B \to M}}{Q_M} = \frac{\bar{\delta}_M - \bar{\delta}_A}{\bar{\delta}_B - \bar{\delta}_A} \quad , \qquad (3)$$

where $f_{M \leftarrow A}$ and $f_{M \leftarrow B}$ denote the fractions of the mixture M originating from the two sources A and B. Using only tracer signatures, Eq. (3) can determine the relative fractions of the two end-members in the mixture, even if all of the relevant fluxes ($q_{A \to M}$, $q_{B \to M}$, and $Q_M$) are unknown.

For many hydrological problems, it would be helpful to know not only how end-members are combined in mixtures, but also how individual end-members are partitioned among their possible fates. That is, it would be helpful to know not only how end-members are *mixed* (as shown at the bottom of Fig. 1), but also how they are *split* into different fluxes (as shown at the left and right sides of Fig. 1). Whereas end-member *mixing* has been widely explored in hydrology, the potential for new insights from end-member *splitting* has been less widely appreciated. What fraction of winter snowmelt becomes winter
streamflow? What fraction becomes summer streamflow? What fraction eventually evaporates, or transpires? Questions



like these require understanding how end-members (such as snowmelt in this example) are split among their potential fates, rather than how they are mixed.

Recent work hints at the potential benefits of an end-member splitting approach. Von Freyberg et al. (2018b) have recently
shown that one can gain new insights into storm runoff generation by expressing the flux of event water in the storm hydrograph (the classic subject of isotope hydrograph separation) as a fraction of total precipitation rather than total streamflow. In our terminology, von Freyberg et al.'s approach *splits* storm rainfall into two fractions: one that becomes "event water" during the current storm, and another that eventually either evapotranspires or is stored in the catchment, to become base flow or "pre-event water" in future hydrologic events. Similarly, Kirchner (2019, Sects. 2.6, 2.7, 3.5, and 4.7)
has shown how tracer data can be used to estimate "forward new water fractions" and "forward transit time distributions", which quantify the fate of current precipitation (rather than the origins of current streamflow, which is the focus of most conventional approaches to transit time estimation). These "forward" new water fractions and transit time distributions quantify how current precipitation is *split among* future streamflows, rather than quantifying how past precipitation events are *mixed in* current streamflow. The underlying concept is not new, dating back at least to Eq. 7 of Niemi (1977) in the
context of transit time distributions. However, it has not been widely recognized that a similar approach can also be applied in end-member mixing analysis, to infer the partitioning of the end-members themselves. Our purpose here is to outline the potential of this approach, which we call end-member splitting.

End-member splitting is based on the observation that (for example) the fraction of end-member A that becomes mixture M
(end-member splitting) is directly related to the fraction of mixture M that is derived from end-member A (end member mixing). These fractions both have the same numerator, the flux $q_{A \to M}$ that flows from A to M; they just have different denominators, $Q_A$ in the first case and $Q_M$ in the second (see Fig. 1). This in turn implies that we can perform end-member splitting by rescaling the results of end-member mixing, through multiplying by the ratio of $Q_M$ to $Q_A$:

$$\eta_{A \to M} = \frac{q_{A \to M}}{Q_A} = \frac{Q_M}{Q_A} \frac{q_{A \to M}}{Q_M} = \frac{Q_M}{Q_A} f_{M \leftarrow A} = \frac{Q_M}{Q_A} \frac{\bar{\delta}_M - \bar{\delta}_B}{\bar{\delta}_A - \bar{\delta}_B} \quad , \tag{4}$$

where $\eta_{A \to M}$ is the proportion of end-member A that eventually becomes mixture M, and $f_{M \leftarrow A}$ is the fraction of mixture M that originated as end-member A. Since all of end-member A must eventually become either part of mixture M or another output (or combination of outputs), here denoted X, we can straightforwardly calculate $\eta_{A \to X}$, the fraction of A that eventually becomes X, by mass balance:

$$\eta_{A \to X} = \frac{q_{A \to X}}{Q_A} = 1 - \eta_{A \to M} = 1 - \frac{Q_M}{Q_A} \frac{\bar{\delta}_M - \bar{\delta}_B}{\bar{\delta}_A - \bar{\delta}_B} \quad . \tag{5}$$

One can also directly calculate the magnitudes of the fluxes connecting each end-member to each output, e.g.,





$$q_{A \to M} = Q_A\, \eta_{A \to M} = Q_M\, f_A = Q_M\, \frac{\bar{\delta}_M - \bar{\delta}_B}{\bar{\delta}_A - \bar{\delta}_B} \quad , \qquad (6)$$

and

$$q_{A \to X} = Q_A\, \eta_{A \to M} = Q_A - Q_M\, \frac{\bar{\delta}_M - \bar{\delta}_B}{\bar{\delta}_A - \bar{\delta}_B} \quad . \qquad (7)$$

We use the symbol $\eta$ to represent how an end-member is partitioned among multiple outputs, to explicitly distinguish it from

the mixing fraction $f$, which represents how a mixture is composed of multiple end-members. We specifically use the
symbol $\eta$ because in thermodynamics it represents efficiency, and $\eta_{A \to M}$ (for example) can be interpreted as the efficiency
with which end-member A is transformed into the mixed output M.

If the un-sampled outputs X and Y can be pooled together (for example, as annual evapotranspiration fluxes), we can

straightforwardly calculate the fractional contributions of each end-member to this pooled output (here denoted XY) as

$$f_{XY \gets A} = \frac{q_{A \to XY}}{Q_{XY}} = \frac{Q_A}{Q_{XY}}\eta_{A \to X} = \frac{Q_A}{Q_{XY}}(1 - \eta_{A \to M}) = \frac{Q_A - Q_M\, \frac{\bar{\delta}_M - \bar{\delta}_B}{\bar{\delta}_A - \bar{\delta}_B}}{Q_A + Q_B - Q_M} \quad . \qquad (8)$$

This calculation requires not only that the fluxes $Q_A$, $Q_B$, and $Q_M$ are known, but that they are known precisely enough that
the mass balance $Q_{XY} = Q_A + Q_B - Q_M$ can be quantified with reasonable accuracy.

Whereas end-member mixing only requires measurements of the volume-weighted tracer composition in the mixture and all
of its potential sources, end-member splitting additionally requires measurements of the water fluxes in the end-members
and mixture(s). Both end-member mixing and end-member splitting analyses should always be accompanied by uncertainty
estimates (quantified via, for example, Gaussian error propagation), to avoid over-interpretation of highly uncertain results.
Gaussian error propagation formulas for the main equations in this paper are presented in the Supplement, and quantities in

the main text and the figures are shown ± standard errors.

Like end-member mixing, end-member splitting can be generalized to more than two sources, if the number of tracers equals
at least the number of sources minus one, and if the tracers are sufficiently uncorrelated with one another. End-member
splitting can also be generalized straightforwardly to any number of mixtures, even using only one tracer if each mixture

combines only two end-members; in the general case, the number of (not-too-correlated) tracers in each mixture must equal
at least the number of end-members minus one.



## 2 Proof-of-concept application

### 2.1 Field site and data

As a proof-of-concept demonstration, here we apply end-member splitting analysis to Campbell and Green's (2019) measurements of $\delta^{18}O$ and $\delta^2H$ at Hubbard Brook Experimental Forest, Watershed 3. Campbell and Green (2019) measured $\delta^{18}O$ and $\delta^2H$ in time-integrated bulk precipitation samples, and instantaneous streamwater grab samples, taken at Watershed 3 approximately every two weeks between October 2006 and June 2010 (Fig. 2); the isotope sampling and analysis procedures are documented in Green et al. (2015). We also used daily precipitation and streamflow measurements for

Watershed 3 compiled from 1958 through 2014 by the USDA Forest Service Northern Research Station (2016a, b).

Watershed 3 is a small (42.4 ha) headwater basin that has served as a hydrologic reference watershed for manipulation experiments conducted in several other nearby watersheds (Bailey et al., 2003). Its soils are well-drained Spodosols with a 3-15 cm thick, highly permeable organic layer at the surface, underlain by glacial drift of highly variable thickness

(averaging roughly 0.5 m, Bailey et al., 2014), which in turn overlies schist and granulite bedrock that is believed to be highly impermeable (Likens, 2013). Ground cover is northern hardwood forest, comprising mainly American beech (Fagus grandifolia Ehrh.), sugar maple (Acer saccharum Marsh.), and yellow birch (Betula alleghaniensis Britt.) (Green et al., 2015), with a growing season extending from June through September (Fahey et al., 2005). Watershed 3 has a humid continental climate, with average monthly temperatures ranging from -8 C in January to 18 C in July (Bailey et al., 2003).

Annual average precipitation was 136 cm $yr^{-1}$ from 1958 through 2014, distributed relatively evenly throughout the year, and annual average streamflow was about 87 cm $yr^{-1}$, implying evapotranspiration losses of roughly 49 cm $yr^{-1}$, or about one-third of average precipitation (USDA Forest Service Northern Research Station, 2016a, b). Approximately 30% of annual precipitation falls as snow, mostly from December through March, reaching an average annual maximum accumulation of 19 cm snow water equivalent (Campbell et al., 2010) and supplying springtime snowmelt pulses in streamflow, which typically

peak in April.

We adjusted Campbell and Green's precipitation isotope values to account for the difference between the mean catchment elevation (642 m; Ali et al., 2015) and the elevation at the precipitation sampler (564 m; Campbell and Green, 2019) assuming an isotopic lapse rate of -0.28 ‰ per 100m for $\delta^{18}O$ (Poague and Chamberlain, 2001) and eight times this amount

(-2.24 ‰ per 100m) for $\delta^2H$. We weighted each precipitation isotope value by the cumulative precipitation that fell during each sampling interval to calculate seasonal volume-weighted averages of $\delta^{18}O$ and $\delta^2H$ in precipitation. To calculate seasonal volume-weighted averages of $\delta^{18}O$ and $\delta^2H$ in streamflow, we weighted each streamflow isotope value by the cumulative streamflow since the previous sample. We calculated uncertainties for all derived quantities using Gaussian error propagation, based on the standard errors of the average water fluxes and the volume-weighted standard errors of the average

isotope ratios, as described in the Supplement. Quantities are reported ± standard errors.




Isotope signatures in Hubbard Brook precipitation exhibit the typical seasonal pattern of temperate mid-latitudes (Fig. 2a): precipitation is isotopically lighter during winter and heavier during summer. There is also considerable sample-to-sample variability, presumably reflecting differences in water sources, atmospheric moisture trajectories, and atmospheric dynamics

between individual precipitation events. The streamwater samples lie slightly above the local meteoric water line (Fig. 2b), suggesting that either the precipitation samples have been slightly affected by evaporative fractionation within the sample collector, or that the streamwater samples have been affected by sub-canopy moisture recycling (Green et al., 2015).

The seasonal cycle in precipitation isotopes is preserved in streamwater at Watershed 3 (somewhat damped and phase-

shifted), whereas the shorter-term fluctuations in precipitation isotopes are almost entirely damped away (Fig. 2a). The strong damping in short-term isotope fluctuations indicates that "event" water from recent precipitation comprises only a small fraction of streamflow, which instead consists mostly of "pre-event" water from many previous precipitation events, thus averaging together their isotopic signatures (Hooper and Shoemaker, 1986; Kirchner, 2003). Over longer time scales, the damping and phase-lagging of the seasonal isotopic cycle directly imply that a fraction of each season's precipitation is

stored in the catchment (as snowpack, soil water, or deeper groundwater, for example), eventually becoming streamflow in future seasons. But how much winter precipitation eventually becomes summer streamflow (for example), and vice versa? How much summer (or winter) precipitation eventually evapotranspires? Quantitative answers to questions like these can shed light on how catchments store and partition water on seasonal time scales.

Our goal is to quantify how precipitation is partitioned between streamflow and evapotranspiration, both within an individual season and between seasons. Figure 3 shows the seasonal cycles in precipitation and streamflow isotopes at Watershed 3, averaged over the entire period of record. Monthly average isotope signatures in precipitation (dark blue symbols in Fig. 3a) reveal two isotopically distinct seasons: a four-month snow-dominated winter (December through March, with isotopically light precipitation), and an eight-month rain-dominated summer (April through November, with isotopically heavy

precipitation). We base our analysis on these two seasons, despite their different lengths, because the results will be most precise if the two inputs are as isotopically distinct as possible. These two seasons coincide with monthly mean air temperatures above and below freezing (gray reference line in Fig. 3f). Here we will refer to either the snowy and rainy seasons, or winter and summer, interchangeably, but neither end-member mixing nor end-member splitting requires the winter season to be snow-dominated.

**2.2 Seasonal origins of summer and winter streamflow**

The damping of the seasonal precipitation isotopic cycle, as seen in Fig. 2a, implies that streamflow during each season must represent a mixture of precipitation from both seasons, potentially spanning multiple years. We can use conventional end-member mixing analysis to straightforwardly estimate how summer and winter precipitation combine to form seasonal





streamflow. Because the two seasons are defined such that they span the entire year, stream discharge in each season must

be derived from a combination of summer and/or winter precipitation:

$$Q_s = q_{P_s \to Q_s} + q_{P_w \to Q_s} \quad , \quad Q_w = q_{P_s \to Q_w} + q_{P_w \to Q_w} \tag{9}$$

where $Q_s$ and $Q_w$ represent the average annual sums of stream discharge during the summer and winter seasons, and (for example) $q_{P_s \to Q_s}$ and $q_{P_w \to Q_s}$ are the average annual fluxes of summer streamflow that originated as summer and winter precipitation, respectively. Equation (9) directly implies that, no matter how the precipitation end-members are defined, they

must jointly account for all the precipitation that could eventually become streamflow (including, potentially, precipitation in multiple previous summers or winters). In other words, streamflow must be composed only of a mixture of the summer and winter precipitation, $P_s$ and $P_w$; there can be no other end members, sampled or not (although obviously streamflow can contain flows from various catchment compartments in which summer and winter precipitation have been stored and mixed). We also assume isotopic mass balance for the water that eventually becomes discharge,

$$Q_s \bar{\delta}_{Q_s} = q_{P_s \to Q_s} \bar{\delta}_{P_s} + q_{P_w \to Q_s} \bar{\delta}_{P_w} \quad \text{and} \quad Q_w \bar{\delta}_{Q_w} = q_{P_s \to Q_w} \bar{\delta}_{P_s} + q_{P_w \to Q_w} \bar{\delta}_{P_w} \quad , \tag{10}$$

where $\bar{\delta}_{Q_s}$, $\bar{\delta}_{Q_w}$, $\bar{\delta}_{P_s}$, and $\bar{\delta}_{P_w}$ are the volume-weighted average isotopic signatures in summer and winter streamflow and precipitation. Equation (10) implies that the precipitation that eventually becomes streamflow does not undergo substantial isotopic fractionation (the effects of which are discussed further in Sect. 3.3). It does not imply that no such fractionation occurs in the water fluxes that are eventually evapotranspired (and in any case, evapotranspiration fluxes are neither sampled

nor directly measured). Combining Eqs. (9) and (10) yields the end-member mixing equations for summer streamflow,

$$f_{Q_s \leftarrow P_s} = \frac{q_{P_s \to Q_s}}{Q_s} = \frac{\bar{\delta}_{Q_s} - \bar{\delta}_{P_w}}{\bar{\delta}_{P_s} - \bar{\delta}_{P_w}} \quad \text{and} \quad f_{Q_s \leftarrow P_w} = \frac{q_{P_w \to Q_s}}{Q_s} = \frac{\bar{\delta}_{Q_s} - \bar{\delta}_{P_s}}{\bar{\delta}_{P_w} - \bar{\delta}_{P_s}} \quad , \tag{11}$$

where $f_{Q_s \leftarrow P_s}$ and $f_{Q_s \leftarrow P_w}$ represent the fractions of summer streamflow that originated as summer and winter precipitation, respectively. An analogous pair of end-member mixing equations can be used to estimate the fractions of winter streamflow that originate as summer and winter precipitation.


As Fig. 4 shows, Eq. (11) and the isotope data from Watershed 3 imply that about 38% of summer (rainy-season) streamflow originates as winter (snowy-season) precipitation, and 62% originates as rainy-season precipitation. They also imply that about 40% of winter (snowy-season) streamflow originates as snowy-season precipitation, and 60% as rainy-season precipitation. These percentages should be assessed in comparison with the proportions of precipitation that originate in the

snowy and rainy seasons. At Watershed 3, the rainy season comprises two-thirds of the year and 70% of total precipitation, as a long-term average. If summer and winter streamflow were derived proportionally from each season's precipitation, each would consist of 70% rainy-season precipitation and 30% snowy-season precipitation. Using these percentages as a reference point, we can quantify how the contributions of summer and winter precipitation to streamflow deviate from their shares of total precipitation, using relationships of the form





$$\Delta f_{Q_s \leftarrow P_s} = \frac{f_{Q_s \leftarrow P_s} - \frac{P_s}{P}}{\frac{P_s}{P}} = \frac{\bar{\delta}_{Q_s} - \bar{\delta}_{P_w}}{\bar{\delta}_{P_s} - \bar{\delta}_{P_w}} \frac{P}{P_s} - 1 \quad \text{and} \quad \Delta f_{Q_s \leftarrow P_w} = \frac{f_{Q_s \leftarrow P_w} - \frac{P_w}{P}}{\frac{P_w}{P}} = -\Delta f_{Q_s \leftarrow P_s} \frac{P_s}{P_w} \quad , \qquad (12)$$

where $\Delta f_{Q_s \leftarrow P_s}$ and $\Delta f_{Q_s \leftarrow P_w}$ are the fractional over- or under-representation of each season's precipitation in summer streamflow. These calculations yield the result that winter precipitation is over-represented by 26% and 32% (and summer precipitation is under-represented by 11% and 14%) in summer and winter streamflow, respectively. The under-representation of summer precipitation in both seasons' streamflow implies that it is over-represented in evapotranspiration

(as examined in Sect. 2.3 below).

More generally, the isotope data from Watershed 3 imply that substantial fractions of streamflow are derived from water that has been stored in the catchment from previous seasons, as either snowpack or groundwater (and, in the case of groundwater, potentially also including water from previous years). Many hydrograph separation studies, including the work of Hooper

and Shoemaker (1986) at Watershed 3, have shown that streamflow is often composed primarily of pre-event water. The results in this section, which can be loosely considered as a seasonal-scale hydrograph separation, extend the previous event-scale findings by showing that even at the seasonal time scale, streamflow is not clearly dominated by current (i.e., same-season) precipitation.

### 2.3 Seasonal origins of evapotranspiration

We can straightforwardly extend the seasonal end-member mixing approach above, to estimate how much evapotranspiration originates as summer vs. winter precipitation. We begin by assuming that the water fluxes satisfy mass balance:

$$P_s + P_w = Q + ET \qquad , \qquad (13)$$

where $P_s$ and $P_w$ represent the average annual sums of precipitation falling in the summer and winter, respectively, $Q$

represents annual average discharge, and $ET$ represents average annual evapotranspiration. Equation (13) assumes that these fluxes are much larger than any other inputs (such as direct surface condensation or groundwater inflows) or outputs (such as groundwater outflow). Equation (13) is also assumed to hold over time scales long enough that changes in catchment storage are trivial compared to the cumulative input and output fluxes. These same assumptions are invoked in hydrometric studies that infer $ET$ from long-term catchment water balances (e.g., Vadeboncoeur et al., 2018). However, such

hydrometric studies cannot reliably estimate the seasonal origins of evapotranspiration, because changes in catchment storage may be substantial on seasonal time scales.



We can straightforwardly apply end-member mixing to the total annual discharge, analogously to the approach used in Eqs. (9)-(11) for discharge during the individual seasons. All discharge must originate as either summer or winter precipitation,

and thus

$$Q = q_{P_s \rightarrow Q} + q_{P_w \rightarrow Q} \quad , \qquad (14)$$

where $q_{P_s \rightarrow Q}$ and $q_{P_w \rightarrow Q}$ are the annual average fluxes that originate as summer and winter precipitation. Isotopic mass balance for the water that eventually becomes discharge implies

$$Q\, \bar{\delta}_Q = q_{P_s \rightarrow Q}\, \bar{\delta}_{P_s} + q_{P_w \rightarrow Q}\, \bar{\delta}_{P_w} \quad , \qquad (15)$$

where $\bar{\delta}_Q$ is the volume-weighted isotopic signature of total annual streamflow. Jointly solving Eqs. (14) and (15) yields the seasonal end-member mixing equations for total annual streamflow,

$$f_{Q \leftarrow P_s} = \frac{q_{P_s \rightarrow Q}}{Q} = \frac{\bar{\delta}_Q - \bar{\delta}_{P_w}}{\bar{\delta}_{P_s} - \bar{\delta}_{P_w}} \quad \text{and} \quad f_{Q \leftarrow P_w} = \frac{q_{P_w \rightarrow Q}}{Q} = \frac{\bar{\delta}_Q - \bar{\delta}_{P_s}}{\bar{\delta}_{P_w} - \bar{\delta}_{P_s}} \quad . \qquad (16)$$

where $f_{Q \leftarrow P_s}$ and $f_{Q \leftarrow P_w}$ represent the fractions of total annual streamflow that originate as summer and winter precipitation, respectively. Using the input data shown in Fig. 4, Eq. (16) yields the result that average annual streamflow is composed of

57±7% rainy-season precipitation and 43±7% snowy-season precipitation.

What does this have to do with evapotranspiration? A consequence of the assumed water balance closure (Eq. 13) is that all precipitation must eventually become either evapotranspiration or discharge, that is,

$$P_s = q_{P_s \rightarrow Q} + q_{P_s \rightarrow ET} \quad , \qquad P_w = q_{P_w \rightarrow Q} + q_{P_w \rightarrow ET} \quad , \qquad (17)$$

where $q_{P_s \rightarrow Q}$ and $q_{P_s \rightarrow ET}$ (for example) represent the average annual fluxes of discharge and streamflow that originate as summer precipitation (potentially including summer precipitation in previous years). Thus summer and winter precipitation that does not eventually become streamflow must contribute to evapotranspiration. Combining Eqs. (13), (16), and (17), one directly obtains the fraction of ET originating as summer precipitation, $f_{ET \leftarrow P_s}$:

$$f_{ET \leftarrow P_s} = \frac{q_{P_s \rightarrow ET}}{ET} = \frac{P_s - q_{P_s \rightarrow Q}}{P_s + P_w - Q} = \frac{P_s - Q\, f_{Q \leftarrow P_s}}{P_s + P_w - Q} = \frac{P_s - Q\, \dfrac{\bar{\delta}_Q - \bar{\delta}_{P_w}}{\bar{\delta}_{P_s} - \bar{\delta}_{P_w}}}{P_s + P_w - Q} \quad . \qquad (18)$$

An analogous expression can be used to estimate $f_{ET \leftarrow P_w}$, the fraction of ET originating as winter precipitation.

As Fig. 4 shows, Eq. (18) implies that evapotranspiration at Watershed 3 is almost entirely (85±15%) derived from rainy-season precipitation, and the fraction derived from snowy-season precipitation is not distinguishable from zero (15±15%). This result is not particularly surprising, for several reasons. First, the rainy season is twice as long as the snowy season, and

accounts for 70% of total annual precipitation. Second, the higher temperatures and vapor pressure deficits that prevail during the summer imply that both surface evaporation rates and potential evapotranspiration rates will be higher during the





rainy season. Third, the growing season of Watershed 3's mixed hardwood forest occurs during the rainy season, implying that transpiration rates during the snowy season should be small. Thus the results of Eq. (18) are biologically and climatologically plausible.


It should be noted that although the lopsided ET source attribution shown in Fig. 4 is not surprising, neither is it intuitively obvious. Intuitively one might assume that since streamflow at Watershed 3 is a mixture of roughly equal fractions of summer and winter precipitation, they should also each comprise roughly half of evapotranspiration. The isotopic mass-balance calculation in Eq. (18) shows that this intuition is wrong, and it also suggests why: annual $ET$ is considerably smaller

than annual $Q$, and winter precipitation is considerably smaller than summer precipitation (partly because the summer is twice as long). Thus winter precipitation can be greatly under-represented in ET while also being roughly half (in fact, less than half) of discharge.

Following the approach in Eq. (12), we can quantify the fractional over- or under-representation of summer and winter
precipitation in total (summer plus winter) streamflow as

$$\Delta f_{Q \leftarrow P_s} = \frac{f_{Q \leftarrow P_s} - \frac{P_s}{P}}{\frac{P_s}{P}} = \frac{\bar{\delta}_Q - \bar{\delta}_{P_w}}{\bar{\delta}_{P_s} - \bar{\delta}_{P_w}} \frac{P}{P_s} - 1 \quad \text{and} \quad \Delta f_{Q \leftarrow P_w} = \frac{f_{Q \leftarrow P_w} - \frac{P_w}{P}}{\frac{P_w}{P}} = -\Delta f_{Q \leftarrow P_s} \frac{P_s}{P_w} \quad , \qquad (19)$$

and the fractional over- or under-representation of summer and winter precipitation in total ET as

$$\Delta f_{ET \leftarrow P_s} = \frac{f_{ET \leftarrow P_s} - \frac{P_s}{P}}{\frac{P_s}{P}} = \frac{P_s - Q \frac{\bar{\delta}_Q - \bar{\delta}_{P_w}}{\bar{\delta}_{P_s} - \bar{\delta}_{P_w}}}{P_s + P_w - Q} \frac{P}{P_s} - 1 \quad \text{and} \quad \Delta f_{ET \leftarrow P_w} = \frac{f_{ET \leftarrow P_w} - \frac{P_w}{P}}{\frac{P_w}{P}} = -\Delta f_{ET \leftarrow P_s} \frac{P_s}{P_w} \quad . \qquad (20)$$

These calculations yield the result that summer precipitation is under-represented by 19% in annual streamflow (summer
precipitation is 70% of annual precipitation but only 61% of annual streamflow, so summer precipitation is under-represented in streamflow by 19%), and winter-precipitation is over-represented by 28%. By contrast, winter precipitation is under-represented in ET by 50% (winter precipitation accounts for 30% of annual precipitation but only 15% of ET, or only about half of ET's share of total precipitation), and summer precipitation is over-represented by 22%.

Finally, it is worth noting that one can infer the average isotopic composition of the unmeasured ET flux straightforwardly by isotope mass balance,

$$\bar{\delta}_{ET} = \frac{P_s \, \bar{\delta}_{P_s} + P_w \, \bar{\delta}_{P_w} - Q \, \bar{\delta}_Q}{P_s + P_w - Q} \quad . \qquad (21)$$

If the associated uncertainties are acceptably small (see error propagation in the Supplement), inferred values of $\bar{\delta}_{ET}$ could be useful in interpreting tree ring records. They could also potentially be useful in quantifying the relative contributions of





evaporation and transpiration to ET at whole-catchment scale, if one can also directly measure the isotopic composition of the evaporation and transpiration fluxes (through soil and xylem sampling, for example).

**2.4 End-member splitting of seasonal precipitation into seasonal discharge and evapotranspiration**

Up to this point we have analyzed evapotranspiration and seasonal discharge as mixtures of summer and winter precipitation. In this section, we analyze the corresponding question of how summer and winter precipitation are partitioned among these

outputs. That is, having addressed the question of where the outputs come from, we now address the mirror-image question of where the inputs go. Mathematically this can be accomplished by re-scaling the end-member mixing results by the ratios of output fluxes to input fluxes, as introduced in Section 1. Consider, for example, the annual average flux $q_{P_s \to Q_s}$ of summer precipitation that becomes summer streamflow. This flux, divided by the annual sum of summer streamflow (the total output flux), yields $f_{Q_s \leftarrow P_s}$, the fraction of summer streamflow that originated as summer precipitation (Eq. 11). But

this same flux, when divided by annual sum of summer precipitation (the total input flux), yields the fraction of summer precipitation that eventually becomes summer streamflow. This fraction, here denoted $\eta_{P_s \to Q_s}$, can therefore be directly calculated from $f_{Q_s \leftarrow P_s}$ by multiplying by the ratio of the output flux to the input flux:

$$\eta_{P_s \to Q_s} = \frac{q_{P_s \to Q_s}}{P_s} = \frac{Q_s}{P_s} \frac{q_{P_s \to Q_s}}{Q_s} = \frac{Q_s}{P_s} f_{Q_s \leftarrow P_s} = \frac{Q_s}{P_s} \frac{\bar{\delta}_{Q_s} - \bar{\delta}_{P_w}}{\bar{\delta}_{P_s} - \bar{\delta}_{P_w}} \quad . \tag{22}$$

Similar relationships can be used to calculate the fraction of summer precipitation that eventually becomes winter

streamflow,

$$\eta_{P_s \to Q_w} = \frac{q_{P_s \to Q_w}}{P_s} = \frac{Q_w}{P_s} \frac{q_{P_s \to Q_w}}{Q_w} = \frac{Q_w}{P_s} f_{Q_w \leftarrow P_s} = \frac{Q_w}{P_s} \frac{\bar{\delta}_{Q_w} - \bar{\delta}_{P_w}}{\bar{\delta}_{P_s} - \bar{\delta}_{P_w}} \quad , \tag{23}$$

the fraction that eventually becomes streamflow in either season,

$$\eta_{P_s \to Q} = \frac{q_{P_s \to Q}}{P_s} = \frac{Q}{P_s} \frac{q_{P_s \to Q}}{Q_w} = \frac{Q}{P_s} f_{Q \leftarrow P_s} = \frac{Q}{P_s} \frac{\bar{\delta}_Q - \bar{\delta}_{P_w}}{\bar{\delta}_{P_s} - \bar{\delta}_{P_w}} \quad , \tag{24}$$

and the fraction that is eventually evapotranspired,

$$\eta_{P_s \to ET} = \frac{q_{P_s \to ET}}{P_s} = \frac{ET}{P_s} f_{ET \leftarrow P_s} = 1 - \eta_{P_s \to Q} = 1 - \frac{Q}{P_s} f_{Q \leftarrow P_s} = 1 - \frac{Q}{P_s} \frac{\bar{\delta}_Q - \bar{\delta}_{P_w}}{\bar{\delta}_{P_s} - \bar{\delta}_{P_w}} \quad . \tag{25}$$

Analogous equations can be used to similarly partition winter precipitation among the same outputs. Intriguingly, Eq. (25) does not require calculating the mass balance $ET = P_s + P_w - Q$; thus one can calculate the fraction of each season's precipitation that is eventually transpired, even if the evapotranspiration rate itself is not well constrained by mass balance.





As Fig. 4 shows, Eqs. (22)-(25) imply that roughly half (44±8%) of rainy-season precipitation is eventually evapotranspired. The remainder is partitioned between summer and winter streamflow in roughly a 2:1 ratio (39±6% and 18±3% of rainy-season precipitation, respectively). By contrast, much less (and perhaps none at all) of snowy-season precipitation (18±18%) is eventually evapotranspired, although the remainder is split between summer and winter streamflow in nearly the same 2:1 ratio (55±13% and 27±6%, respectively) as the rainy-season precipitation is partitioned. This 2:1 ratio is perhaps

unsurprising, because the summer season is twice as long as the winter season, and summer streamflow is 68% of total streamflow, but it implies significant carryover of water from each season to the next.

  Figure 4 illustrates how end-member mixing and end-member splitting yield different (but complementary) perspectives on the catchment water balance. Only about half of rainy-season precipitation is eventually evapotranspired, but nearly all

evapotranspiration originates as rainy-season precipitation. The two proportions are different but not inconsistent, for the simple reason that rainy-season precipitation is much greater than annual evapotranspiration. Likewise, both rainy-season and snowy-season precipitation are split between rainy- and snowy-season streamflow in a 2:1 ratio, but streamflow during both seasons originates from roughly equal proportions of snowy- and rainy-season precipitation. Again the proportions are different but not inconsistent, since total rainfall and total streamflow are both greater during the rainy season than during the

snowy season.

  As with the mixing fractions derived in Sects. 2.2 and 2.3, we can also express end-member splitting proportions in terms of how much the possible fates of precipitation are over- or under-represented, relative to their flow-proportional share of total precipitation. For example, from Fig. 4 one can see that roughly one-third of summer precipitation ultimately becomes

summer streamflow; is this more, or less, than one would expect if precipitation were split among all of its fates proportionally to their total fluxes? If precipitation were split proportionally among summer streamflow, winter streamflow, and evapotranspiration, and if summer and winter precipitation were both split by the same proportions, then the proportion of precipitation that ultimately became summer streamflow would be $\frac{Q_s}{P} = 0.44$. This provides a reference point for comparing the actual end-member splitting result of $\eta_{P_s \rightarrow Q_s}$=39±6%:

$$\Delta\eta_{P_s \rightarrow Q_s} = \frac{\Delta\eta_{P_s \rightarrow Q_s} - \frac{Q_s}{P}}{\frac{Q_s}{P}} = \frac{P}{P_s}f_{Q_s \leftarrow P_s} - 1 = \frac{P}{P_s}\frac{\bar{\delta}_{Q_s} - \bar{\delta}_{P_w}}{\bar{\delta}_{P_s} - \bar{\delta}_{P_w}} - 1 = \Delta f_{Q_s \leftarrow P_s} \quad . \tag{26}$$

It may seem strange that $\Delta\eta_{P_s \rightarrow Q_s}$, the fractional over- or under-representation of summer streamflow as a fate for summer precipitation, is numerically equal to $\Delta f_{Q_s \leftarrow P_s}$, the fractional over- or under-representation of summer precipitation in summer streamflow. This is particularly so, given that the end-member splitting proportion $\eta_{P_s \rightarrow Q_s}$ (Eq. 22) is substantially different from the end-member mixing fraction $f_{Q_s \leftarrow P_s}$ (Eq. 11), and the two metrics are compared to two different reference





points ($\frac{Q_s}{P}$ for $\eta_{P_s \to Q_s}$ and $\frac{P_s}{P}$ for $f_{Q_s \leftarrow P_s}$).   However, because the ratio between these reference points is $\frac{Q_s}{P_s}$ and the ratio

between $\eta_{P_s \to Q_s}$ and $f_{Q_s \leftarrow P_s}$ is also $\frac{Q_s}{P_s}$, it follows mathematically that $\Delta\eta_{P_s \to Q_s} = \Delta f_{Q_s \leftarrow P_s}$.   The same phenomenon holds for

the under- or over-representation of winter streamflow as a fate of summer precipitation, for which an appropriate point of

reference is $\frac{Q_w}{P}$:,

$$\Delta\eta_{P_s \to Q_w} = \frac{\Delta\eta_{P_s \to Q_w} - \frac{Q_w}{P}}{\frac{Q_w}{P}} = \frac{P}{P_s} f_{Q_w \leftarrow P_s} - 1 = \frac{P}{P_s} \frac{\bar{\delta}_{Q_w} - \bar{\delta}_{P_w}}{\bar{\delta}_{P_s} - \bar{\delta}_{P_w}} - 1 = \Delta f_{Q_w \leftarrow P_s} \quad , \tag{27}$$

and the under- or over-representation of annual streamflow as a fate of summer precipitation, for which an appropriate point

of reference is $\frac{Q}{P}$:

$$\Delta\eta_{P_s \to Q} = \frac{\Delta\eta_{P_s \to Q} - \frac{Q}{P}}{\frac{Q}{P}} = \frac{P}{P_s} f_{Q \leftarrow P_s} - 1 = \frac{P}{P_s} \frac{\bar{\delta}_Q - \bar{\delta}_{P_w}}{\bar{\delta}_{P_s} - \bar{\delta}_{P_w}} - 1 = \Delta f_{Q \leftarrow P_s} \quad , \tag{28}$$

and the under- or over-representation of evapotranspiration as a fate of summer precipitation, for which an appropriate point

of reference is $\frac{ET}{P}$:


$$\Delta\eta_{P_s \to ET} = \frac{\Delta\eta_{P_s \to ET} - \frac{ET}{P}}{\frac{ET}{P}} = \frac{P}{P_s} f_{ET \leftarrow P_s} - 1 = \frac{P}{P_s} \frac{P_s - Q\frac{\bar{\delta}_Q - \bar{\delta}_{P_w}}{\bar{\delta}_{P_s} - \bar{\delta}_{P_w}}}{P_s + P_w - Q} - 1 = \Delta f_{ET \leftarrow P_s} \quad . \tag{29}$$

Naturally, one can also write analogous expressions for the corresponding fractions of winter precipitation.  Using Eqs. (26)-
(29) and the information in Fig. 4, one can calculate that the fractions of summer precipitation going to summer and winter
streamflow are 11% and 14% less, and the fraction going to ET is 22% greater, than their proportional shares of total
precipitation.  By contrast, the fractions of winter precipitation going to summer and winter streamflow are 26% and 31%
greater, and the fraction going to ET is 50% less, than their proportional shares of total precipitation.  These percentages do
not balance because they are percentages of different quantities (the proportions of total outflows).

Stepping back from these details, however, the most striking result of the end-member splitting analysis is that 18% of rainy-
season precipitation (or 160 mm yr⁻¹), and 55% of snowy-season precipitation (or 219 mm yr⁻¹), leaves the catchment as
streamflow during a different season than the one that it fell in.  This reinforces the point that there must be significant inter-
seasonal water storage at the catchment scale.  The annual snowpack clearly represents a significant inter-seasonal storage of
winter precipitation, because much of its melt takes place in April, which is during the rainy season.  Annual peak snowpack
storage is roughly 190 mm of snow water equivalent, which equals roughly 70% of average winter precipitation, and



apparently a substantial fraction of this crosses into the rainy season to become streamflow (for example, during the
snowmelt pulse in April), but only a small fraction is evapotranspired.

End-member splitting calculations are based on mass balances, and therefore must be applied to long-term average fluxes,
for which mass balances can be assumed to be reasonably precise. The calculations outlined in this section further assume
that the sampled precipitation and streamflow are representative of the snowy and rainy seasons. Of course, the inputs to any
such calculation will inevitably be based on finite sets of samples and measurements, which may deviate somewhat from the
(unknown) long-term averages. How sensitive are the results to the specific periods that we analyzed? How much
uncertainty would be introduced if the available records were even more limited? To get some idea, we extracted three
individual water years, each running from December to November (and thus each including one snowy season and one rainy
season), from the isotope and water flux time series. We then repeated the end-member splitting analysis using only data
from each individual water year (daily precipitation and discharge fluxes, and a total of roughly 24 biweekly isotope
measurements in precipitation and streamflow). The results are shown in Fig. 5, which also compares end-member splitting
proportions obtained from oxygen-18 (shown by circles) with those obtained from deuterium (shown by diamonds). Figure
5 shows that when one uses shorter data sets (light blue symbols) the resulting uncertainties are bigger, as expected, but the
error bars overlap with the estimates derived from the entire data set (dark blue symbols, based on all available isotope data,
and long-term average water fluxes). These results demonstrate that the small-sample estimates are realistic approximations
(within their standard errors) of the values that would be derived from the more complete data set.

### 2.5 Partitioning of seasonal precipitation into monthly discharges

Because we have only one tracer in practice (we nominally have both oxygen-18 and deuterium, but they are largely
redundant with one another), end-member mixing can quantify the fractional contributions from only two sources (such as
summer and winter precipitation) in each mixture (such as summer and winter streamflow). There is, however, no
mathematical limit to the number of different mixtures that such end-member mixing calculations could be applied to.
(There may be a logical limit, of course; it would make little sense to express streamflow on each individual day as a mixture
of summer and winter precipitation, given the wide variability in precipitation isotopes from one storm to the next.) Because
there is no mathematical limit on the number of different mixtures, in the context of end-member splitting there is no
mathematical limit on the number of different fates that each source can be partitioned among. The only constraint is that
the outputs must jointly account for all of the input (i.e., all of the precipitation must go somewhere), and we must have
tracer and water flux measurements for all-but-one of them. In most practical cases, the unmeasured output will be
evapotranspiration (or will be _called_ evapotranspiration, although it will formally be the sum of all unmeasured fluxes).

Here we illustrate this approach by splitting summer and winter precipitation among each month's streamflow, instead of just
summer and winter streamflow. The monthly end-member mixing equations are of the form,





$$f_{Q_i \leftarrow P_s} = \frac{q_{P_s \to Q_i}}{Q_i} = \frac{\bar{\delta}_{Q_i} - \bar{\delta}_{P_w}}{\bar{\delta}_{P_s} - \bar{\delta}_{P_w}} \quad , \tag{30}$$

where $Q_i$ is the monthly discharge in month $i$. The corresponding end-member splitting equations, derived by the logic of Eq. (4), are


$$\eta_{P_s \to Q_i} = \frac{q_{P_s \to Q_i}}{P_s} = \frac{Q_i}{P_s} f_{Q_i \leftarrow P_s} = \frac{Q_i}{P_s} \frac{\bar{\delta}_{Q_i} - \bar{\delta}_{P_w}}{\bar{\delta}_{P_s} - \bar{\delta}_{P_w}} \quad . \tag{31}$$

Analogous equations can be written for the winter precipitation end-member.

The results of this analysis are shown in Fig. 6. Although monthly precipitation rates are roughly equal throughout the year, monthly discharge rates show a distinct snowmelt-driven peak in April and distinct low flows attributable to

evapotranspiration in July, August, and September (Fig. 6a). Monthly end-member mixing (Eq. 30) shows that the mixing fraction $f_{Q_i \leftarrow P_s}$ of summer precipitation in streamflow reaches a minimum of 34% during the spring discharge peak and increases throughout the growing season, peaking at 88% in August (Fig. 6b). The partitioning $\eta_{P_s \to Q_i}$ of summer precipitation among monthly streamflows, however, shows a very different pattern, peaking during spring snowmelt (when the fraction of summer precipitation in streamflow is lowest), and reaching a minimum during the growing season (when the

fraction of summer precipitation in streamflow is highest; Fig. 6c).

This relationship arises because, as Eq. 31 shows, the "forward" partitioning fractions $\eta_{P_s \to Q_i}$ of precipitation (Fig. 6c) are proportional to the "backward" mixing fractions $f_{Q_i \leftarrow P_s}$ (Fig. 6b), which vary by less than a factor of three, multiplied by the monthly discharges $Q_i$ (Fig. 6a), which vary by nearly a factor of nine. Because $Q_i$ is more variable than $f_{Q_i \leftarrow P_s}$, variations

in the "forward" partitioning fractions $\eta_{P_s \to Q_i}$ largely reflect variations in $Q_i$. For example, between April and August the percentage of rainy-season precipitation in streamflow increases from 34 to 88 percent (a factor of 2.5), but the total discharge flux decreases from 205 to 26 mm month[-1] (a factor of nearly 8). Thus although rainy-season precipitation makes up of a greater fraction of rainy-season precipitation in August than in April, August streamflow accounts for a much smaller fraction of rainy-season precipitation than April streamflow does. The same principle also holds for the "forward"

partitioning fractions $\eta_{P_w \to Q_i}$ of winter precipitation, but in this case it is less evident because the seasonal patterns in $Q_i$ and the "backward" mixing fractions $f_{Q_i \leftarrow P_w}$ of winter precipitation generally reinforce, rather than offset, one another. Unsurprisingly, the forward partitioning fractions $\eta_{P_w \to Q_i}$ of winter precipitation among monthly discharges reach their peak during spring snowmelt and their minimum during summer low flows.





The forward partitioning fractions $\eta_{P_s \to Q_i}$ of summer precipitation reach a second peak in late autumn, after the end of the

growing season but before substantial snowfall (Fig. 6c).  During this period, interception and transpiration losses are

relatively small, as one can see from the rise in stream discharge from September through November despite nearly constant

monthly precipitation totals (Fig. 6a).  Thus late autumn streamflows are relatively high.  Because those streamflows also

contain large mixing fractions $f_{Q_i \leftarrow P_s}$ of summer precipitation (Fig. 6b), they result in a peak in the end-member splits of

summer precipitation $\eta_{P_s \to Q_i}$ (Fig. 6c).  Somewhat surprisingly, the partitioning fractions $\eta_{P_w \to Q_i}$ of winter precipitation also

rise somewhat in late autumn, even though the winter season ended more than six months ago (Fig. 6d), and precipitation

does not acquire its winter isotopic signature again until December.  This rise in the late autumn occurs because snowy-

season precipitation still makes up roughly 15% of streamflow (Fig. 6b), presumably reflecting long-term subsurface storage

mobilized by increased infiltration of autumn rainfall after the growing season ends.


In any case, the most striking feature of Figure 6 is that it indicates that substantial export of rainy-season precipitation

occurs just as the snowy season is ending and the rainy season is beginning.  This could result from the big April snowmelt

pulse mobilizing groundwater that was stored through the winter.  Alternatively, it could result from the snowmelt pulse

saturating shallow soil layers and causing large fractions of April rainfall to reach the stream.  The fraction of summer

precipitation in April streamflow is 34±11%, or 69±23 mm month[-1] out of an average April streamflow of 205±5 mm

month[-1].  This 69±23 mm month[-1] must consist of April precipitation, or precipitation from previous summers.  If the 69±23

mm month[-1] were composed entirely of April precipitation, it would account for about 70% of average April precipitation

(106±5 mm month[-1]).  Thus these results do not require that large quantities of summer precipitation must have overwintered

as groundwater, but they also do not exclude that possibility.

**2.6 End-member splitting of growing-season and dormant-season precipitation**

In the analysis presented above in Sects. 2.2-2.5, we separated the year into a rainy season and a snowy season, to maximize

the isotopic difference between the two precipitation end-members.  Other precipitation seasons, which are less optimal from

an isotopic separation standpoint, are also possible.  It could be of biological interest, for example, to separate the year into

the growing season (June-September) and the dormant season (October-May).  The analysis proceeds exactly as described in

Eqs. (9)-(29), except now "summer" and "winter" correspond to the growing and dormant seasons, respectively.  As Figs. 3c-

3d show, the precipitation isotopes in the growing and dormant seasons are less distinct than those in the rainy and snowy

seasons, for the simple reason that the dormant season includes both rain-dominated months (October-November and April-

May) and snow-dominated months (December-March).  As a consequence, mixing fractions and end-member splits

calculated from the growing-season and dormant-season end-members will inevitably have larger uncertainties than those

calculated from the rainy- and snowy-season end-members.  Nonetheless, as Fig. 7 shows, one can still draw useful

inferences from such end-member mixing and splitting calculations.  From Fig. 7 one can see that nearly all (84±21%) of



dormant-season streamflow originates from dormant-season precipitation, and the contribution from growing-season precipitation is zero within error (16±21%). Conversely, roughly half (45±19%) of growing-season streamflow originates from dormant-season precipitation, and the other half (55±19%) originates from growing-season precipitation.

Evapotranspiration appears to be mostly (60±35%) derived from growing-season precipitation, with a smaller contribution (40±35%) from dormant-season precipitation, but the uncertainties are large enough that many other mixing fractions are also possible. End-member splitting shows that a large fraction (72±18%) of dormant-season precipitation eventually becomes dormant-season streamflow, with a small but well-defined fraction (6±2%) eventually becoming growing-season streamflow and a larger but uncertain fraction (22±19%) potentially being evapotranspired. Conversely, a large but

uncertain fraction (62±36%) of growing-season precipitation is eventually evapotranspired, with a small but well-defined fraction (14±5%) eventually becoming growing-season streamflow and a small and highly uncertain fraction (24±32%) becoming dormant-season streamflow.

It is noteworthy that, in Fig. 7, dormant-season precipitation makes up about half (45±19%) of growing-season discharge,

and nearly all (79±20%) of total annual discharge, but probably less than half (40±35%) of evapotranspiration. Conversely, growing-season precipitation probably makes up the bulk (60±35%) of evapotranspiration, but only a small fraction (21±20%) of total annual discharge. This example illustrates how an isotopic separation between "blue water" and "green water" (the so-called "two water worlds" phenomenon) could arise through unsurprising contrasts between the proportions of winter and summer precipitation that eventually become evapotranspiration vs. streamflow. We emphasize that this analysis

makes no specific inference about how, mechanistically, such a separation occurs. Importantly, however, this isotopic separation does not require that "blue water" and "green water" are sourced from physically distinct storages. In particular, it does not require a separation between "bound waters" that primarily supply ET and "mobile waters" that primarily supply streamflow (Brooks et al., 2010; Good et al., 2015), although it also does not rule this out. Instead, our analysis shows that isotopic evidence of apparent "two water worlds" requires only that evapotranspiration rates vary seasonally, and that

catchments do not store enough water to average out the isotopic differences between summer and winter precipitation when those waters become ET or streamflow. These conditions are likely to be met in many catchments.

As a further thought experiment, we can ask how snowy- and rainy-season precipitation contribute to – and are partitioned among – dormant- and growing-season streamflow. Here we make use of the fact that the analyses derived above do not

require us to use the same seasons to characterize precipitation and streamflow. Thus we can repeat the same analysis that is outlined in Eqs. (9)-(29), using "summer" to refer to growing-season (June-September) streamflow but rainy-season (April-November) precipitation, and "winter" to refer to dormant-season (October-May) streamflow but snowy-season (December-March) precipitation. Naturally, one must keep in mind the different lengths of these seasons, as well as their sometimes substantial differences in water fluxes, when interpreting the results.






The results of this analysis are shown in Fig. 8. Just as in Fig. 4, evapotranspiration is derived almost entirely (85±15%) from rainy-season precipitation, and relatively little, or almost not at all, (15±15%) from snowy-season precipitation. These results are identical to those obtained in Sect. 2.3 because, in our analysis, ET is not (and cannot be) differentiated by season (unless we have measurements of the ET fluxes themselves, or of their isotopic signatures). Thus we can distinguish the

seasonal origins of ET fluxes, but not the seasons in which those ET fluxes occur. Figure 8 shows that growing-season streamflow is derived in roughly a 4:1 ratio from rainy-season and snowy-season precipitation (79±8% and 21±8%, respectively), whereas dormant-season streamflow is derived from nearly equal contributions from the two seasons (58±9% and 42±9%, respectively). Roughly half of rainy-season precipitation eventually evapotranspires; a roughly equal amount (46±7%) becomes dormant-season streamflow, and a small but well-constrained fraction (10±1%) becomes growing-season

streamflow. It is striking that this 10% fraction of rainy-season precipitation makes up the dominant fraction (79±8%) of growing-season streamflow, but this simply reflects the fact that rainy-season precipitation is nearly eight times larger than growing-season streamflow. This is partly due to substantial evapotranspiration losses during the growing season, and also due to the fact that the growing season is only half as long as the rainy season. It may seem striking that about four times as much rainy-season precipitation becomes dormant-season streamflow as becomes growing-season streamflow. However

this is not as surprising as it first might seem, given that half of the rainy season overlaps with the dormant season (April-May and October-November), and that the other half of the rainy season (i.e., the growing season) is marked by substantial evapotranspiration losses and very low streamflows. The great majority (77±16%) of snowy-season precipitation becomes dormant-season streamflow, which is unsurprising because both the snowy season and the snowmelt period are contained within the dormant season. Thus, not only is evapotranspiration almost entirely sourced from rainy-season precipitation over

the three summers for which measurements are available, it also appears that relatively little snowy-season precipitation could compensate for ecosystem water shortages during summer droughts, because most snowy-season precipitation becomes streamflow in the dormant season. A small but well-defined fraction (6±2%) of snowy-season precipitation becomes growing-season streamflow, and a small and indefinite fraction (17±18%) evapotranspires. It is noteworthy that about one-fifth of growing-season streamflow is derived from snowy-season precipitation, despite the fact that growing

season begins two months after the snowy season ends. Thus this fraction of snowy-season precipitation (roughly 25 mm yr$^{-1}$) must be stored in the subsurface for at least several months, before becoming growing-season streamflow.

### 2.7 Comparison with sine-wave fitting and young water fractions

Sections 2.2-2.4 and 2.6 draw inferences concerning intra- and inter-seasonal storage and transport by comparing seasonal isotopic variations in precipitation and streamflow. Seasonal isotope cycles have been used to infer timescales of catchment

storage for at least two decades, since at least the work of DeWalle et al. (1997). The damping of seasonal isotopic cycles has recently been shown to quantify the average fraction of streamflow that is younger than approximately 2-3 months, even in spatially heterogeneous and nonstationary catchments (Kirchner, 2016a, b). Figure 9 shows volume-weighted seasonal sinusoidal cycles fitted to the deuterium time series. The ratio between the volume-weighted seasonal cycle amplitudes in



streamflow and precipitation ($A_Q^*$ and $A_P^*$, respectively) yields the volume-weighted young water fraction $F_{yw}^* = A_Q^*/A_P^*$, the

proportion (by volume) of streamflow that is younger than roughly 2-3 months. (Here we follow von Freyberg et al. (2018a) in using an asterisk to denote volume-weighted quantities.) The cycles in Fig. 9 imply a volume-weighted young water fraction $F_{yw}^*$ of 0.45±0.09, which is broadly comparable to the $f_{Q_w \leftarrow P_w} = 40±9\%$ of snowy-season Q that originates as snowy-season P and the $f_{Q_s \leftarrow P_s} = 55±19\%$ of growing-season Q that originates as growing-season P (both 4-month seasons), and also consistent with the $f_{Q_s \leftarrow P_s} = 62±9\%$ of rainy-season Q that originates as rainy-season P and the $f_{Q_w \leftarrow P_w} = 84±21\%$ of

dormant-season Q that originates as dormant-season P (both 8-month seasons).

Following the approach of Eq. (4), we can multiply the volume-weighted young water fraction by the ratio between the average streamflow and average discharge to obtain the young water fraction of precipitation $^P F_{yw}^* = F_{yw}^* \bar{Q}/\bar{P}$, the average fraction (by volume) of precipitation that leaves the catchment as streamflow within 2-3 months. The cycles in Fig. 9 imply

that the young water fraction of precipitation $^P F_{yw}^*$ is 0.29±0.06, which can be compared to the $\eta_{P_w \to Q_w} = 27±6\%$ of snowy-season precipitation that becomes snowy-season streamflow, and the $\eta_{P_s \to Q_s} = 14±5\%$ of growing-season precipitation that becomes growing-season streamflow (both 4-month seasons), or the $\eta_{P_s \to Q_s} = 39±6\%$ of rainy-season precipitation that becomes rainy-season streamflow and the $\eta_{P_w \to Q_w} = 72±18\%$ of dormant-season precipitation that becomes dormant-season streamflow (both 8-month seasons). Precise mathematical comparisons are not possible, because these 4- and 8-month

seasons are not directly comparable to the 2-3 month time scale of the young water fractions $F_{yw}^*$ and $^P F_{yw}^*$, and also because these young water fractions are annual averages whereas the $f$'s and $\eta$'s pertain to individual seasons. Nonetheless, all of these lines of evidence imply that significant fractions of streamflow must originate from precipitation in previous seasons, and conversely that significant fractions of precipitation become streamflow in future seasons. This in turn implies significant water storage within the catchment, either as snowpack or as groundwater.

**2.8 Comparison with new water fractions estimated by ensemble hydrograph separation**

Another approach for quantifying timescales of storage and transport using isotopic tracers is ensemble hydrograph separation. Ensemble hydrograph separation uses the regression slope between tracer fluctuations in streamwater and precipitation to quantify the "new water fraction", the average fraction of streamflow that is "new" since the previous precipitation sample (Kirchner, 2019). Thus, in this case, because the precipitation isotopes are averaged over a roughly

two-week sampling interval, the new water fraction quantifies the fraction of streamflow that is younger than about two weeks. This biweekly new water fraction, $^Q F_{new}$, can be estimated from the regression slope parameter $\beta$ in the linear regression equation,

$$y_j = \beta x_j + \alpha + \varepsilon_j \quad \text{with} \quad y_j = \delta_{Q_j} - \delta_{Q_{j-1}} \quad \text{and} \quad x_j = \bar{\delta}_{P_j} - \delta_{Q_{j-1}} \quad , \tag{32}$$





where $\overline{\delta}_{P_j}$ and $\delta_{Q_j}$ are the isotope signatures in precipitation and streamflow, respectively, in the $j^{th}$ sampling interval (and

where the overbar on $\overline{\delta}_{P_j}$ indicates that it is an average over that interval).  If many sampling intervals have no precipitation, one must account for the number of intervals with precipitation, as a fraction of the total (see Kirchner, 2019 for details), but here we can overlook this because nearly every two-week interval at Hubbard Brook has precipitation.  Weighting the regression in Eq. (32) by discharge yields the volume-weighted new water fraction of streamflow, $^{Q}F_{new}^{*}$.  Uncertainty estimates for $^{Q}F_{new}^{*}$ and similar volume-weighted quantities should take account of the reduced degrees of freedom that

result from the uneven weighting, as described in Eq. (19) of Kirchner (2019).

Following the approach of Eq. (4), we can multiply $^{Q}F_{new}^{*}$ by the ratio of mean discharge to mean precipitation to obtain the volume-weighted new water fraction of precipitation $^{P}F_{new}^{*}$, the fraction of precipitation that, on average, leaves the catchment as streamflow within the sampling interval (in this case, two weeks):

$$^{P}F_{new}^{*} \;=\; {^{Q}F_{new}^{*}}\, \frac{\overline{Q}}{\overline{P}} \quad . \tag{33}$$

In the language of Sect. 1, Eq. (33) *splits* the precipitation end-member into two fractions: the average fraction that leaves as streamflow within the sampling interval ( $^{P}F_{new}^{*}$), and the average fraction that doesn't ($1 - {^{P}F_{new}^{*}}$).  For this reason, $^{P}F_{new}^{*}$ can also be termed a "forward" new water fraction because it divides precipitation into two different future fates.  Likewise $^{Q}F_{new}^{*}$ can be termed a "backward" new water fraction because it divides streamflow according to its origins as precipitation

in the recent or distant past.  In contrast to end-member mixing and end-member splitting, this approach is based on correlations between tracer fluctuations in streamflow and precipitation, rather than mass balances.  Thus it can be applied even if the underlying tracer time series are incomplete.

Applying this approach to the Hubbard Brook record, and using the total discharge in each sampling interval as weights, we

estimate the volume-weighted biweekly new water fraction of discharge $^{Q}F_{new}^{*}$ as 8.3±1.9%, and the corresponding volume-weighted biweekly new water fraction of precipitation $^{P}F_{new}^{*}$ as 5.3±1.2%.  These results mean that, on average, about five percent of precipitation leaves the catchment as streamflow in the following two weeks, and this makes up about eight percent of streamflow.

One can also apply this regression approach to subsets of the data, highlighting time periods or catchment conditions of particular interest (Kirchner, 2019).  For comparison with the results presented in Sects. 2.4 and 2.6 above, I divided the time series into four seasons: the four-month snowy season (December-March), the four-month growing season (June-September), and the two-month spring and fall seasons in between (April-May and October-November, respectively).  The volume-weighted regressions for these four seasons (Fig. 10) show that tracer fluctuations in precipitation and streamflow





are weakly correlated during the snowy season (Fig. 10a), much more strongly correlated in the spring (Fig. 10b), and
        correlated to an intermediate degree during the growing season and the fall (Fig. 10c-d).  The volume-weighted biweekly
        new water fraction of discharge $^Q F_{new}^*$ is zero within error (2.2±3.3%) during the snowy season (Fig. 10a), even though at
        the four-month seasonal timescale (Fig. 4), roughly half of snowy-season streamflow originates as snowy-season
        precipitation.  Considered together, these results would seem to imply that almost all winter precipitation is stored in the
catchment for at least two weeks (as either snowpack or subsurface storage), effectively decoupling precipitation and
        streamflow on that timescale, but roughly half eventually melts or seeps out to streams sometime during the winter.  During
        the growing season (Fig. 10c), the volume-weighted biweekly new water fraction of discharge $^Q F_{new}^*$ is 10.6±2.8%.  This is
        broadly consistent with the observation that, on a seasonal timescale, about half of growing-season streamflow originates as
        growing-season precipitation (Fig. 7), although an exact equivalence is difficult to draw because the fraction of "new" water
in streamflow declines over time following each event.  During the fall the biweekly new water fraction is similar
        (11.6±3.4%), but during the spring it is distinctly higher (22.0±7.8%), presumably due to more saturated catchment
        conditions.

        The biweekly new water fractions of precipitation $^P F_{new}^*$ yield further insights.  The biweekly new water fraction of
precipitation is markedly higher during the spring (31.1±11.1%), reflecting greater transmission of new water to streamflow
        under wet catchment conditions.  Very little precipitation is transmitted to streamflow on a two-week time frame during
        either the snowy season (1.5±2.2%) or the growing season (2.7±0.7%), reflecting the fact that there is relatively little
        streamflow of any kind during those periods.  In the snowy season this is due to snowpack storage; in the growing season it
        is due to evapotranspiration.  The essential difference between the two is that the snowpack episodically melts, with the
result that about one-fourth of snowy-season precipitation eventually becomes snowy-season streamflow (Figure 4), whereas
        the evapotranspired water is lost for good, with the result that only about 10% of growing-season precipitation eventually
        becomes growing-season streamflow (Figure 7).

        Figure 11 shows the same ensemble hydrograph separation approach, applied separately to each month of the year.  The
volume-weighted biweekly new water fraction of discharge $^Q F_{new}^*$ is lowest in January and February (when temperatures at
        Hubbard Brook are the coldest), and peaks during snowmelt in April.  The rest of the year it hovers around 10 percent.  The
        volume-weighted biweekly new water fraction of precipitation $^P F_{new}^*$ is zero within error from January through March, then
        abruptly rises to 43±25% during April, declines to 2% or less throughout the growing season from June through September,
        then rises to 5-9% until the end of the year.  Here again we see the effects of winter freezing and summer evapotranspiration
in limiting streamflow (as well as recent contributions of precipitation to it).  We also see the effects of catchment wetness
        during snowmelt facilitating the transmission of large fractions of recent precipitation to streamflow, as well as the increase
        in precipitation reaching the stream from October through December, following the cessation of the growing season.  This





analysis provides striking evidence that during about half of the year, in mid-summer and mid-winter, nearly no precipitation reaches the stream during the first two weeks after it falls. More generally, this analysis also demonstrates that ensemble

hydrograph separation can yield useful insights into the partitioning of precipitation into prompt and more distant streamflow, even based on biweekly tracer data. Furthermore, this analysis shows that new water fractions of precipitation can be combined with end-member splitting analyses, to gain insight into evapotranspiration and subsurface storage as controls on how much recent precipitation reaches streams.

### 3 Assumptions, limitations, and applications

**3.1 Fundamental assumptions**

Many of the assumptions underlying end-member splitting are the same as those that underlie end-member mixing. End-member mixing requires, fundamentally, that there are only two end-members (if we have one tracer), or $n+1$ end-members (if we have $n$ non-redundant tracers), that contribute to the measured mixture(s). (More crucially, end-member mixing requires that these are the only end-members *in the real world*, not just the only end-members in your theory, your model, or

your sampling program!) This assumption is broadly met by our two end-members, because precipitation is the ultimate source of catchment streamflow and evapotranspiration (assuming other inputs such as groundwater inflows, condensation, or fog deposition are trivial by comparison), and because we have divided annual precipitation into two seasons, without gaps or overlaps.

End-member mixing also requires that the tracer signatures of the end-members and mixture(s) have been measured without bias. This assumption is broadly met, in our case, by measuring the volume-weighted average isotope signatures of precipitation and streamflow, and measuring them for long enough that carryover effects at the beginning and end of the period are likely to be small. However, one must also be aware of possible isotopic fractionation in the precipitation sampler itself. It is also possible that an unbiased sample of precipitation could nonetheless be a biased sample of the precipitation

that actually becomes streamflow. If, for example, lower-intensity precipitation events tend to be isotopically heavier (Dansgaard, 1964) and more likely to be lost to canopy interception, an unbiased sample of precipitation will be isotopically heavier than the precipitation that eventually flows through the catchment and becomes streamflow. This in turn would lead to an underestimate of summer precipitation (and an underestimate of winter precipitation) as contributors to streamflow.

Lastly, end-member mixing requires that the tracer signatures of the fluxes connecting the end-members to the mixture(s) are not substantially altered by fractionation (i.e., tracer-selective addition or removal of water). For example, although evaporation fluxes are likely to be strongly fractionated, if the waters that are left behind eventually evaporate completely (as may often occur during canopy interception, for example; Allen et al., 2017), the remaining precipitation that eventually becomes streamflow may not be substantially fractionated. Streamwater at Hubbard Brook lies close to the local meteoric





water line (Fig. 2b), suggesting that any such fractionation effects are likely to be small.  Nonetheless, in Sect. 3.3 below, we
      quantify how different types of fractionation would affect our analysis.

      In addition to the assumptions outlined above for end-member mixing, end-member splitting additionally requires that the
      sampled mixture(s) represent all of the outputs from the system except one, and that the water fluxes in these all-but-one
outputs, as well as the end-members, can be quantified with reasonable accuracy.  One can see from Eqs. (4), (22)-(24), and
      (31) that uncertainties in these water fluxes will propagate proportionally through to uncertainties in the end-member
      splitting fractions.  In addition, calculating the end-member mixing fractions of evapotranspiration fluxes (Eqs. 8 and 18)
      requires that the other inputs and outputs are known precisely enough that ET can be calculated with sufficient accuracy by
      mass balance.  Our proof-of-concept at demonstration Hubbard Brook is facilitated not just by the availability of isotope
data, but also by a reliable long-term catchment water balance.

      **3.2 Sensitivity to errors in mass fluxes**

      End-member mixing calculations are not based on mass flux measurements and therefore are independent of errors in mass
      fluxes (except to the extent that they are needed to accurately estimate volume-weighted tracer signatures for the end-
      members and mixtures).  End-member splitting calculations, on the other hand, require mass flux measurements and thus are
potentially vulnerable to errors in them.  We can straightforwardly calculate the sensitivity of these calculations to mass flux
      errors by (for example) differentiating Eq. (22) by its two component fluxes:

$$\frac{\partial \eta_{P_s \to Q_s}}{\partial Q_s} = \frac{1}{P_s} \frac{\bar{\delta}_{Q_s} - \bar{\delta}_{P_w}}{\bar{\delta}_{P_s} - \bar{\delta}_{P_w}} = \frac{\eta_{P_s \to Q_s}}{Q_s} \quad or \quad \frac{\partial \eta_{P_s \to Q_s}}{\eta_{P_s \to Q_s}} = \frac{\partial Q_s}{Q_s} \tag{34}$$

      and

$$\frac{\partial \eta_{P_s \to Q_s}}{\partial P_s} = -\frac{Q_s}{P_s{}^2} \frac{\bar{\delta}_{Q_s} - \bar{\delta}_{P_w}}{\bar{\delta}_{P_s} - \bar{\delta}_{P_w}} = -\frac{\eta_{P_s \to Q_s}}{P_s} \quad or \quad \frac{\partial \eta_{P_s \to Q_s}}{\eta_{P_s \to Q_s}} = -\frac{\partial P_s}{P_s} \quad . \tag{35}$$

Equations (34)-(35) show that an $x$ percent overestimate in $Q_s$ would lead, all else equal, in an $x$ percent overestimate in the
      end-member splitting fraction $\eta_{P_s \to Q_s}$, and an $x$ percent overestimate in $P_s$ would lead, all else equal, in an $x$ percent
      underestimate in $\eta_{P_s \to Q_s}$.  Equation (35) assumes that $x$ is small; if that is not the case, one can directly simulate the effect of
      large errors in $P_s$ by solving Eq. (22) for a range of $P_s$ values.

We can similarly differentiate Eq. (18) by its three component fluxes to quantify how flux measurement errors would affect
      estimates of the fraction of ET originating as summer precipitation, $f_{ET \gets P_s}$:

$$\frac{\partial f_{ET \gets P_s}}{\partial P_s} = \frac{1 - f_{ET \gets P_s}}{ET} \quad , \quad \frac{\partial f_{ET \gets P_s}}{\partial P_w} = \frac{-f_{ET \gets P_s}}{ET} \quad , \text{ and } \quad \frac{\partial f_{ET \gets P_s}}{\partial Q} = \frac{f_{Q \gets P_s} - f_{ET \gets P_s}}{ET} \quad . \tag{36}$$

Figure 12a shows how errors in the water fluxes $P_s$, $P_w$, and $Q$ at Watershed 3 would alter the estimates of $f_{ET \leftarrow P_s}$ and $\eta_{P_s \rightarrow ET}$ shown in Fig. 4. As one can see from Fig. 12a, $f_{ET \leftarrow P_s}$ is least sensitive to errors in $P_s$ (solid light blue curve); this is because

$P_s$ appears in both the numerator and denominator of Eq. (18), with mostly offsetting effects. Although $Q$ also appears in both the numerator and denominator, in the numerator it is multiplied by $f_{Q \leftarrow P_s}$ so errors in $Q$ will not have such cleanly offsetting effects (dashed light blue curve). Errors in $P_w$ (dotted light blue curve) are the most consequential because $P_w$ appears only in the denominator of Eq. (18). Readers will note that sufficiently severe flux measurement errors can lead to calculated values of $f_{ET \leftarrow P_s}$ that exceed 1; this nonphysical result can arise when the water fluxes and tracer signatures in Eq.

(18) become sufficiently inconsistent with one another.

### 3.3 Potential effects of isotopic fractionation

End-member splitting, just like end-member mixing, is potentially vulnerable to the effects of isotopic fractionation. If, for example, a fraction of precipitation evaporates from the rainfall collector, the remaining water, which will be sampled and analyzed, will be isotopically heavier than the precipitation that it is supposed to represent. Alternatively, if the precipitation

samples themselves are not isotopically fractionated, but the precipitation that enters the catchment is fractionated before it becomes streamflow, then the sampled precipitation will be isotopically lighter than the precipitation that it is supposed to represent (i.e., the precipitation that eventually becomes part of streamflow). How much the precipitation that reaches the stream is fractionated will depend, not only on how much evaporates and on ambient temperature and humidity under which that evaporation occurs, but also on how much the evaporating waters are mixed with (or separated from) the waters that are

left behind (Brooks et al., 2010; Sprenger et al., 2016). To the extent that the evaporating waters are separated from those that ultimately reach the stream, their isotopic fractionation will not be reflected in the streamflow isotope signature. An example of such a process is canopy interception; if the intercepted precipitation mostly evaporates after the rain has stopped, and evaporates completely, it leaves no isotopic signal in the water that reaches the stream (Gat and Tzur, 1967; Allen et al., 2017). Alternatively, if the evaporation flux comes from a well-mixed pool that also supplies streamflow, that

streamflow will bear the isotopic fingerprint of evaporative fractionation, with streamflow falling below the local meteoric water line on a dual-isotope plot. In any case, a benefit of using stream water to infer the seasonal origins of evapotranspired waters is that fractionation effects should be much smaller than they would be in sampled xylem or soil water, for which evaporation effects must be compensated to infer their seasonal origins (Benettin et al., 2018; Bowen et al., 2018; Allen et al., 2019).


One can straightforwardly estimate how isotopic fractionation would affect end-member mixing and splitting fractions, by differentiating the corresponding equations by the corresponding input isotope values. For example, we can differentiate Eq. (11) by its three isotopic inputs to quantify how isotopic fractionation could alter estimates of $f_{Q_s \leftarrow P_s}$, the fraction of summer streamflow that originates as summer precipitation:





$$\frac{\partial f_{Q_s\leftarrow P_s}}{\partial \bar{\delta}_{P_s}} = \frac{-f_{Q_s\leftarrow P_s}}{\bar{\delta}_{P_s} - \bar{\delta}_{P_w}} \quad , \quad \frac{\partial f_{Q_s\leftarrow P_s}}{\partial \bar{\delta}_{P_w}} = \frac{f_{Q_s\leftarrow P_s} - 1}{\bar{\delta}_{P_s} - \bar{\delta}_{P_w}} \quad , \quad \text{and} \quad \frac{\partial f_{Q_s\leftarrow P_s}}{\partial \bar{\delta}_{Q_s}} = \frac{1}{\bar{\delta}_{P_s} - \bar{\delta}_{P_w}} \quad , \tag{37}$$

where $f_{Q_s\leftarrow P_s} = \left(\bar{\delta}_{Q_s} - \bar{\delta}_{P_w}\right)\left(\bar{\delta}_{P_s} - \bar{\delta}_{P_w}\right)^{-1}$. The fraction of summer precipitation that eventually becomes summer streamflow, $\eta_{P_s\rightarrow Q_s}$, equals $f_{Q_s\leftarrow P_s}$ rescaled by $Q_s/P_s$, the ratio of summer streamflow to summer precipitation (Eq. 22), so the effects of isotopic fractionation on $\eta_{P_s\rightarrow Q_s}$ are likewise proportional to those derived directly above for $f_{Q_s\leftarrow P_s}$,

$$\frac{\partial \eta_{P_s\rightarrow Q_s}}{\partial \bar{\delta}_{P_s}} = \frac{-\eta_{P_s\rightarrow Q_s}}{\bar{\delta}_{P_s} - \bar{\delta}_{P_w}} \quad , \quad \frac{\partial \eta_{P_s\rightarrow Q_s}}{\partial \bar{\delta}_{P_w}} = \frac{\eta_{P_s\rightarrow Q_s} - Q_s/P_s}{\bar{\delta}_{P_s} - \bar{\delta}_{P_w}} \quad , \quad \text{and} \quad \frac{\partial \eta_{P_s\rightarrow Q_s}}{\partial \bar{\delta}_{Q_s}} = \frac{Q_s/P_s}{\bar{\delta}_{P_s} - \bar{\delta}_{P_w}} \quad . \tag{38}$$

As another example, we can differentiate Eq. (18) by its three isotopic inputs to quantify how isotopic fractionation could alter estimates of $f_{ET\leftarrow P_s}$, the fraction of evapotranspiration that originates as summer precipitation:

$$\frac{\partial f_{ET\leftarrow P_s}}{\partial \bar{\delta}_{P_s}} = \frac{Q}{ET}\frac{f_{Q\leftarrow P_s}}{\bar{\delta}_{P_s} - \bar{\delta}_{P_w}} \quad , \quad \frac{\partial f_{ET\leftarrow P_s}}{\partial \bar{\delta}_{P_w}} = \frac{Q}{ET}\frac{1 - f_{Q\leftarrow P_s}}{\bar{\delta}_{P_s} - \bar{\delta}_{P_w}} \quad , \quad \text{and} \quad \frac{\partial f_{ET\leftarrow P_s}}{\partial \bar{\delta}_{Q}} = \frac{Q}{ET}\frac{-1}{\bar{\delta}_{P_s} - \bar{\delta}_{P_w}} \quad , \tag{39}$$

where $f_{Q\leftarrow P_s} = \left(\bar{\delta}_Q - \bar{\delta}_{P_w}\right)\left(\bar{\delta}_{P_s} - \bar{\delta}_{P_w}\right)^{-1}$. Rescaling $f_{ET\leftarrow P_s}$ by $ET/P_s$, the ratio of evapotranspiration to summer precipitation, yields $\eta_{P_s\rightarrow ET}$, the fraction of summer precipitation that eventually evapotranspires (Eq. 25), we can calculate the effects of isotopic fractionation on $\eta_{P_s\rightarrow ET}$ by rescaling Eq. (39) by the same ratio:

$$\frac{\partial \eta_{P_s\rightarrow ET}}{\partial \bar{\delta}_{P_s}} = \frac{Q}{P_s}\frac{f_{Q\leftarrow P_s}}{\bar{\delta}_{P_s} - \bar{\delta}_{P_w}} \quad , \quad \frac{\partial \eta_{P_s\rightarrow ET}}{\partial \bar{\delta}_{P_w}} = \frac{Q}{P_s}\frac{1 - f_{Q\leftarrow P_s}}{\bar{\delta}_{P_s} - \bar{\delta}_{P_w}} \quad , \quad \text{and} \quad \frac{\partial \eta_{P_s\rightarrow ET}}{\partial \bar{\delta}_{Q}} = \frac{Q}{P_s}\frac{-1}{\bar{\delta}_{P_s} - \bar{\delta}_{P_w}} \quad . \tag{40}$$

Equations (37)-(40) show that, perhaps counterintuitively, if both summer and winter precipitation are fractionated in the same direction, their effects reinforce one another rather than tending to cancel each other out; their terms have the same signs in each of the four equations. For example, an overestimate of $\bar{\delta}_{P_s}$ in Eq. (37) will lead to an underestimate of $f_{Q_s\leftarrow P_s}$, because a larger $\bar{\delta}_{P_s}$ will increase the denominator of $f_{Q_s\leftarrow P_s}$ (see Eq. 11). For example, an overestimate of $\bar{\delta}_{P_w}$ in Eq. (37) will also lead to an underestimate of $f_{Q_s\leftarrow P_s}$, because the numerator of $f_{Q_s\leftarrow P_s}$ will always be smaller than the denominator (since the fraction $f$ must be less than 1), so a larger $\bar{\delta}_{P_w}$ will shrink the numerator of $f_{Q_s\leftarrow P_s}$ more than the denominator in percentage terms.

Figure 12 demonstrates how calculations of $f_{Q_s\leftarrow P_s}$, $f_{ET\leftarrow P_s}$, $\eta_{P_s\rightarrow Q_s}$, and $\eta_{P_s\rightarrow ET}$ would be affected by errors in the mass fluxes and isotope signatures that they use as inputs. Figures 12b and 12c show that errors in $\bar{\delta}_{P_s}$ (solid lines) and $\bar{\delta}_{P_w}$ (dotted lines) reinforce, rather than offset, one another, but that they both would tend to be counteracted by errors in $\bar{\delta}_{Q}$ (dashed lines), assuming that these errors all have the same sign. Figure 12 is based on input values from Fig. 4; for other input values the results would differ in detail, but we expect the overall patterns to be similar.





### 3.4 Potential applications

These methods may provide new insight into how climate change could affect terrestrial ecosystems and water resources. Climate change projections typically involve precipitation increases or decreases in specific seasons, and the tools presented here provide empirical insights into how different seasons' precipitation is partitioned into evapotranspiration or streamflow. At Hubbard Brook Watershed 3, for example, only a small fraction of snowy-season precipitation is evapotranspired (Fig. 4), and a large fraction of evapotranspiration is derived from precipitation that falls during the growing season itself (Fig. 7). These results suggest that tree-ring cellulose is likely to record the isotopic signatures of summer precipitation, rather than those of mean annual precipitation. These results also suggest that forest growth at Hubbard Brook is likely to be sensitive to changes in growing-season precipitation, but less sensitive to changes in winter snowfall. By contrast, roughly half of growing-season streamflow at Watershed 3 originates as precipitation outside of the growing season (Fig. 7), suggesting that summer streamflow could be strongly affected by changes in precipitation in other seasons.

Hypotheses such as these could be tested using isotope records that encompass multiple years with contrasting climates. We could, for example, separate such a long-term record into years with above-average and below-average winter precipitation (or growing-season rainfall). We could then examine how the seasonal partitioning of precipitation, and the seasonal origins of streamflow and evapotranspiration, differed between these different sets of years. If, for example, evapotranspiration fluxes in drier summers are accompanied by smaller contributions from summer precipitation and greater contributions from winter precipitation (smaller $f_{\text{ET}\leftarrow P_s}$ and larger $f_{\text{ET}\leftarrow P_w}$), then winter precipitation may be able to buffer the effects of shifts in summer precipitation on forest growth. Conversely, the lack of such a compensatory response would suggest greater vulnerability of forest growth to changes in summer precipitation. Through such analyses (of which one is underway), we can transition from asking "which seasons' water do ecosystems use?" to asking "which seasons' water do they depend on?"

End-member splitting may also help in illuminating hydrological transport, storage, and mixing processes. For example, if substantial fractions of summer precipitation become summer streamflow despite widespread soil-moisture deficits throughout the catchment (which is not the case at Hubbard Brook), this would indicate that summer precipitation can bypass the soil via preferential flow, contrary to the common model representation of soils as well-mixed "buckets". Such a scenario could explain why trees throughout much of Switzerland were recently found to be using winter precipitation in mid-summer, despite enough summer precipitation having fallen to saturate soils to their median rooting depths (Allen et al., 2019). This example also points to the potential of combining end-member splitting analysis with direct isotopic sampling of xylem water and soil water; such an analysis is now underway using data from a network of Swiss catchments.

The relative amounts of precipitation becoming same-season streamflow or ET, versus "crossing over" to become streamflow or ET in other seasons, also provides a constraint on the shape of the transit time distribution, both of the


precipitation that becomes streamflow and of the precipitation that evapotranspires. End-member splitting may also be helpful for model calibration, validation, and testing, because it provides different information than is provided by

hydrometric input/output data. Unlike direct tests against isotopic time series, end-member splitting analysis provides a "fingerprint" or "signature" of catchment behavior for models to be tested against, an approach that will often have greater diagnostic power (Kirchner et al., 1996). End-member splitting also provides spatially and temporally integrated information, in contrast to point measurements of xylem and soil water, which cannot be readily generalized to the scales of most hydrologic models. Furthermore, because end-member splitting analysis can be performed with relatively short weekly

or biweekly time series, it can potentially be applied in a wide range of sites where only low-frequency isotopic data are available, rather than the few sites where direct model calibration and testing against isotope time series would be feasible.

The analyses presented in Sect. 2, as well as the potential applications outlined in this section, have focused on the coupling of precipitation to streamflow and evapotranspiration within and between seasons. In temperate climates and continental

interiors, such analyses are facilitated by the strong seasonal cycle that is typically found in the isotopic composition of precipitation. All of the approaches presented here require that precipitation can be separated into two seasons that are isotopically distinct. This will not be possible in all cases. Exceptions include coastal or tropical sites lacking strong seasonality in precipitation isotopes, and Mediterranean climates in which almost all precipitation falls within a single season.


Such cases where precipitation isotope seasonality is weak or absent present intractable problems for seasonally oriented analyses, but also present opportunities for analyses based on isotopic differences between other groupings of precipitation events. In field settings spanning large elevation gradients, one could potentially use the isotopic variation in precipitation with altitude (the "altitude effect"; Dansgaard, 1954, 1964; Siegenthaler and Oeschger, 1980), within an end-member

splitting framework, to contrast the fates of precipitation falling in the higher versus the lower parts of a river basin. Alternatively, one could potentially make use of the fact that low-intensity precipitation is often isotopically heavier than high-intensity precipitation, due to greater isotopic fractionation of raindrops as they fall (the "amount effect"; Dansgaard, 1964). Where the contrast between low-intensity and high-intensity storms is the dominant source of variability in precipitation isotopes (e.g., in some tropical regions; Jasechko and Taylor, 2015), end-member splitting analysis could be

used to contrast the fates of low-intensity and high-intensity precipitation, providing new insight into transport, storage, and runoff generation at the catchment scale. As an extreme example of contrasting storm intensities, on could potentially use tropical cyclones and all other precipitation as the two end members, because tropical cyclones are isotopically much lighter than any other tropical precipitation (Lawrence and Gedzelman, 1996).



**4 Concluding remarks**

We make no particular claim for the novelty of the approach we have outlined here, since it represents a conceptually straightforward combination of end-member mixing and isotope mass balance methods, both of which are well established. End-member splitting is nonetheless noteworthy because it represents a different perspective. It invites questions that are seldom asked, such as "where does precipitation go?" (rather than "where did streamflow come from?"), and provides a framework for answering them. Such questions have previously been approached through simulation models (e.g., Benettin

et al., 2015; Benettin et al., 2017), but end-member splitting provides a model-independent way to answer them directly from data.

The analyses presented in Sect. 2 above serve both as a worked example showing how end-member splitting can be applied in practice, and as a proof-of-concept study that illustrates its potential. The techniques outlined in Sect. 2 can be used to

determine the seasonal origins of streamflow (Sect. 2.2) and evapotranspiration (Sect. 2.3), as well as the seasonal partitioning of precipitation into evapotranspiration and streamflow (Sect. 2.4). We also show that one can infer how the seasonal origins of streamflow shift from month to month, and conversely how precipitation is partitioned among monthly streamflows (Sect. 2.5).

Here we have analyzed Hubbard Brook Watershed 3 as a test case. The results illustrate how end-member mixing and splitting yield different insights, which together give a more complete picture of catchment behavior. At Watershed 3, for example, almost all evapotranspiration is derived from rainy-season precipitation, but only about half of rainy-season precipitation eventually transpires (Fig. 4). One sixth of rainy-season precipitation is eventually discharged during the snowy season, but this accounts for half of snowy-season streamflow (Fig. 4). Only about 10% of growing-season

precipitation becomes discharge during the growing season, but this accounts for nearly half of growing-season streamflow (Fig. 7). The other half of growing-season streamflow is derived from just 7% of dormant-season precipitation (Fig. 7). The largest discharges of rainy-season precipitation occur during snowmelt, when rainy-season precipitation makes up the smallest fraction of streamflow; conversely, the smallest discharges of rainy-season precipitation occur during the growing season, when it makes up the largest fraction of streamflow (Fig. 6). In all the cases shown here (Figs. 4, 7, and 8), a

substantial fraction of each season's streamflow originates as precipitation in other seasons. These results therefore imply substantial inter-seasonal catchment storage, in either snowpacks or groundwaters.

**Code and data availability**

An R script that performs the calculations described in this paper will be deposited in an open-access archive and the DOI will be supplied with the final published paper. The source data used in this paper are available from the cited references.



**Author contributions**

JWK and STA jointly developed the end-member splitting approach. JWK performed the analysis presented here and drafted the paper. Both authors discussed all aspects of the work and jointly edited the manuscript.

**Competing interests**

The authors declare that they have no conflict of interest.

**Acknowledgments**

We thank the Hubbard Brook Ecosystem Study, and particularly Mark Green and John Campbell, for making the data that we used in our analysis publicly available.

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





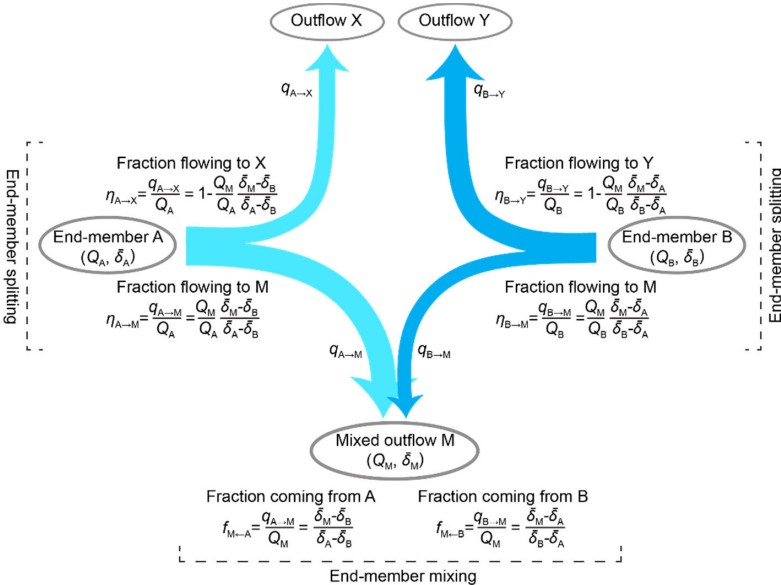

**Figure 1. Schematic illustration of end-member mixing and end-member splitting. Two end-members, A and B, contribute to a mixed outflow M, and to two other outflows, denoted X and Y, respectively. The fluxes between the end-members and outflows are denoted $q_{A\rightarrow M}$, $q_{A\rightarrow X}$, $q_{B\rightarrow M}$, and $q_{B\rightarrow Y}$; these are assumed to not be directly measurable. Conventional end-member mixing, as shown at the bottom of the figure, can be used to calculate the fractions of the two end-members in the mixture using only their volume-weighted average tracer signatures ($\overline{\delta}_A$, $\overline{\delta}_B$, and $\overline{\delta}_M$). If one also knows the water fluxes in the mixed outflow and one or both end-members, one can use end-member splitting, as shown on the left and right sides of the figure, to quantify how the end-members are partitioned between the mixture M and their other outflows X and Y.**


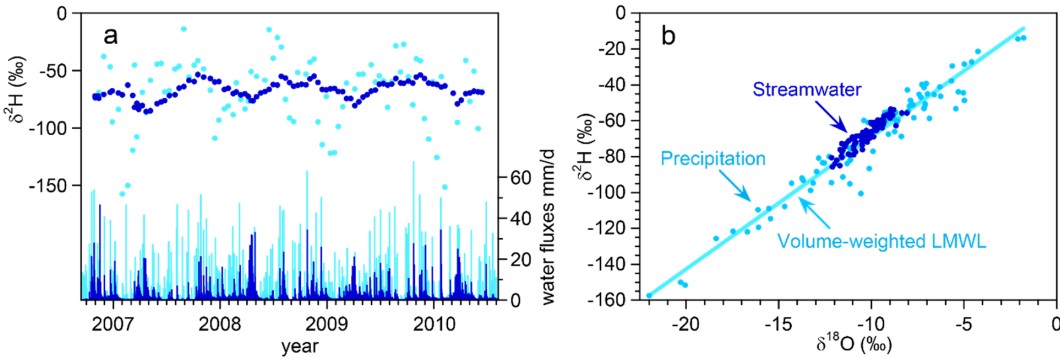

**Figure 2. (a) Time series of daily water fluxes and bi-weekly deuterium values in streamwater (dark blue) and precipitation (light blue) at Watershed 3, Hubbard Brook Experimental Forest (data of Campbell and Green, 2019). (b) Dual-isotope plot showing local meteoric water line computed by volume-weighted regression ($\delta^2H = 4.74\pm2.26 + (7.37\pm0.22)\ \delta^{18}O$). Streamwater lies slightly above the local meteoric water line, on average (lc-excess = 2.91±0.27 ‰, mean±standard error), possibly suggesting slight evaporative fractionation of precipitation within the sample collector, or potential fractionation of streamwater by sub-canopy moisture recycling (Green et al., 2015).**

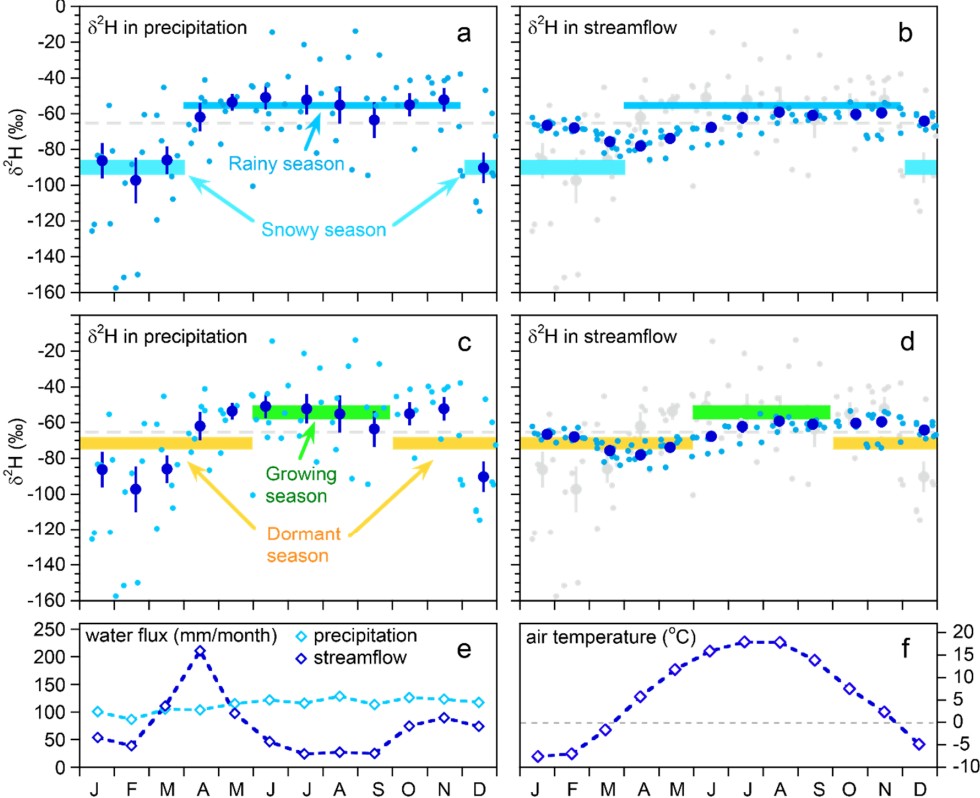

**Figure 3. Seasonal variation in deuterium ratios in bulk samples of precipitation (left-hand plots: a,c) and grab samples of**
**streamflow (right-hand plots: b,d) from 2006 through 2010 at Hubbard Brook Watershed 3. Diamonds in panel (e) are monthly water fluxes averaged over 1958-2014, showing distinct effects of snowmelt in March through May, and evapotranspiration in June through September. Diamonds in panel (f) are monthly mean air temperatures relative to gray reference line of 0 °C. Light blue dots in panels (a-d) show individual samples, with 3 or 4 years of sampling overlapped, depending on month. Dark blue dots show monthly volume-weighted means; error bars show standard errors where these are larger than plotting symbols. Gray**
**dashed line shows the volume-weighted mean for all precipitation. Horizontal bars show seasonal volume-weighted precipitation means ± standard errors, using two different definitions of seasons. The upper plots (a,b) show seasons defined by the break in isotopic composition between months in which precipitation is predominantly rain (April-November) and predominantly snow (December-March). Defining the seasons in this way maximizes the isotopic difference between them. The next two plots (c,d) show the same underlying isotope measurements, but with averages defined for the growing season (June-September) and the**
**dormant season (October-May). These seasons are isotopically less distinct than the rainy/snowy seasons, because the dormant season overlaps the isotopic shifts between November-December and March-April. The seasonal precipitation means are copied in the right-hand plots (along with the individual precipitation values themselves, in gray), for comparison with the streamflow isotope measurements. Streamflow separation into rainy-season vs. snowy-season precipitation sources is more precise, because these seasonal precipitation sources are more distinct, in comparison to growing-season vs. dormant-season precipitation sources.**




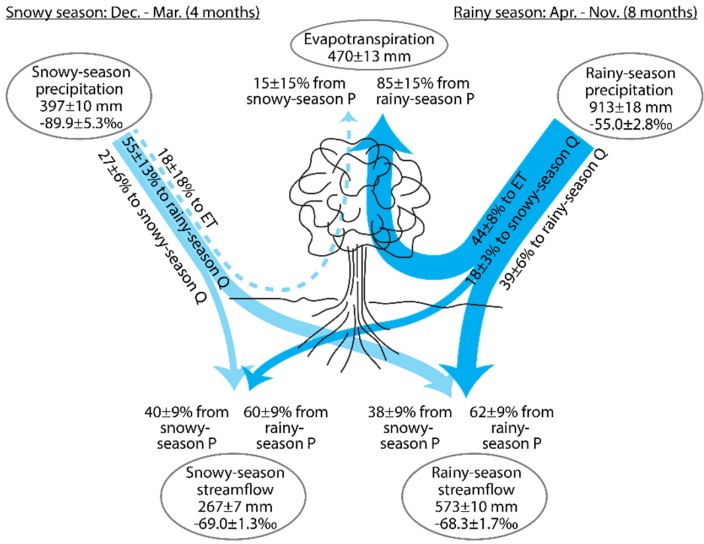

**Figure 4. Partitioning of precipitation (P) into streamflow (Q) and evapotranspiration (ET) during the snow-dominated season (December-March) and the rain-dominated season (April-November), inferred from annual water fluxes and volume-weighted $\delta^2$H at Hubbard Brook Watershed 3. Essentially all evapotranspiration is derived from rainy-season precipitation. Roughly half of rainy-season precipitation eventually evapotranspires, about one third eventually becomes rainy-season streamflow, and about one-sixth eventually becomes snowy-season streamflow. Only about one fourth of snowy-season precipitation becomes snowy-season streamflow, with about half becoming rainy-season streamflow and perhaps one fifth being lost to evaporation and transpiration. Roughly half of each season's streamflow is derived from the other season's precipitation, implying substantial inter-seasonal storage in snowpacks or groundwaters. All quantities are shown ± standard errors. Widths of lines are approximately proportional to water fluxes. Fluxes within one standard error of zero are shown by dashed lines. Percentages may not add to 100 due to rounding.**





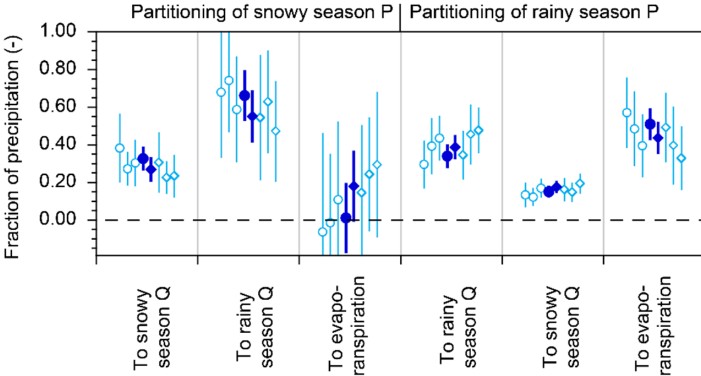

**Figure 5.** Seasonal partitioning of precipitation (P) into streamflow (Q) and evapotranspiration (ET), estimated from δ[18]O
(circles) and δ[2]H (diamonds) from individual water years. Solid symbols show results using all available isotope measurements
and long-term averages of P and Q water fluxes. Open symbols show results using only isotope and water flux measurements
collected during individual water years (2007 through 2009, from left to right). Water years are defined from December through
the following November, thus including one snowy season and the following rainy season. Seasonal partitioning estimates derived
from δ[18]O and δ[2]H generally agree within their standard errors, as do estimates derived from individual years of data (open
symbols). Unsurprisingly, estimates derived from individual years have larger uncertainties than those derived from all available
data.





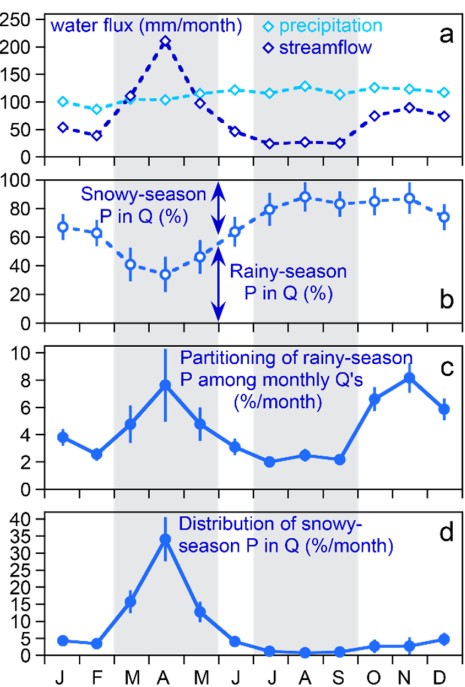

**Figure x6. Patterns in monthly average precipitation and streamflow fluxes (a), isotope hydrograph separations of rainy- and snowy-season precipitation in monthly streamflows (b), and distributions of rainy-season (c) and snowy-season (d) precipitation in streamflow (fraction of precipitation leaving as streamflow in each month). Proportions in (c) and (d) do not sum to 100% because they do not include evapotranspiration losses (which are 8% and 48% of snowy-season and rainy-season precipitation, respectively). Average precipitation fluxes vary little from month to month, whereas average streamflow fluxes show clear high flows resulting from snowmelt from March through May and clear low flows attributable to evapotranspiration losses from July through September (a). Both intervals are marked by gray shading. Monthly isotope hydrograph separations (b) show larger fractions of snowy-season precipitation in streamflow during the snowmelt period, followed by a steadily growing fraction of rainy-season precipitation that reaches a peak of nearly 90% in August. However, much more rainy-season precipitation becomes streamflow during snowmelt (c), when its fractional contribution to streamflow is lowest (b), than during late summer, when its fractional contribution to streamflow is relatively high (b,c). This occurs because monthly total streamflow is much higher during snowmelt than during the high-ET conditions of late summer. A relatively large proportion of rainy-season precipitation also becomes streamflow in October through December, as monthly total streamflow recovers after the end of the summer ET peak. The proportion of snowy-season precipitation becoming streamflow (d) unsurprisingly peaks in during peak snowmelt, when monthly streamflow is highest and the fractional contribution of snowy-season precipitation to that streamflow is likewise high.**




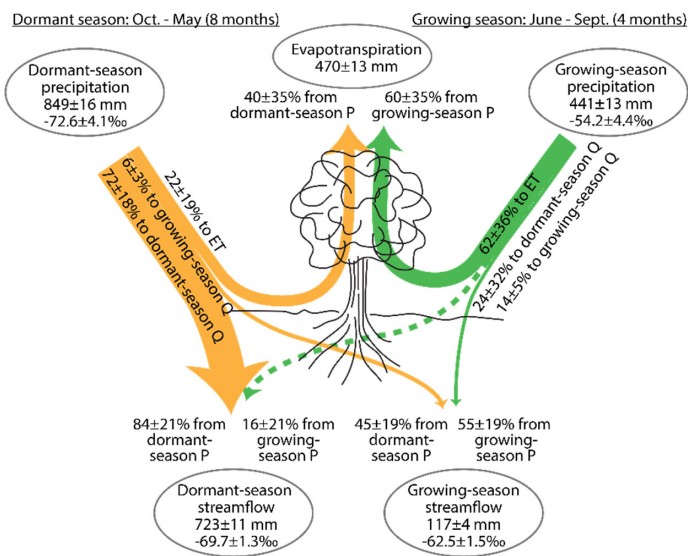

**Figure 7. Partitioning of precipitation (P) into streamflow (Q) and evapotranspiration (ET) during the dormant season (October-May) and the growing season (June-September), inferred from annual water fluxes and volume-weighted δ²H at Hubbard Brook Watershed 3. These two precipitation seasons are less isotopically distinct than the rainy/snowy seasons (see Fig. 3), so the propagated uncertainties are correspondingly larger than those shown in Fig. 4. Evapotranspiration is mostly derived from growing-season precipitation, with a smaller fraction coming from dormant-season precipitation, but both percentages are highly uncertain. Most growing-season precipitation is eventually evapotranspired, with a small but well-defined fraction eventually becoming growing-season streamflow. Roughly half of growing-season streamflow is derived from a small but well-defined fraction of dormant-season precipitation. Most of the rest of dormant-season precipitation eventually becomes dormant-season streamflow, and about one-fifth may evapotranspire (although this is highly unertain). All quantities are shown ± standard errors. Widths of lines are approximately proportional to water fluxes. Fluxes within one standard error of zero are shown by dashed lines.**




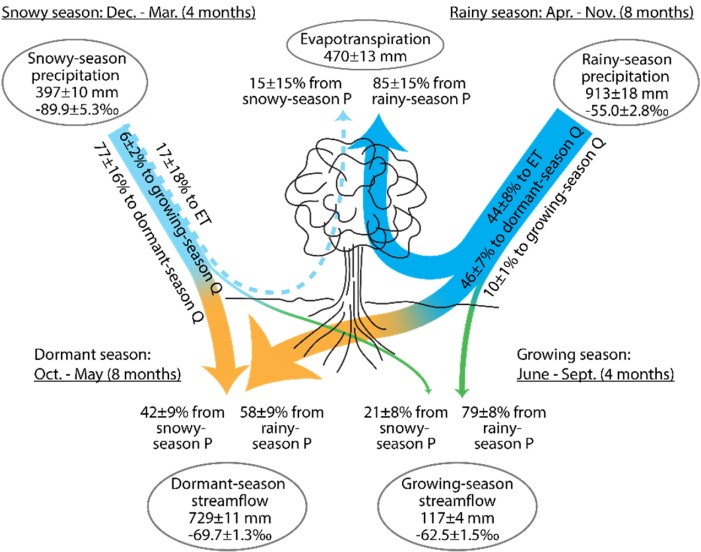

**Figure 8. Partitioning of snowy-season (December-March) and rainy-season (April-October) precipitation (P) into evapotranspiration (ET) and streamflow (Q) during the dormant season (October-May) and the growing season (June-September), inferred from annual water fluxes and volume-weighted δ²H at Hubbard Brook Watershed 3. About half of rainy-season precipitation eventually evapotranspires, and this accounts for almost all the annual evapotranspiration flux; the contribution from snowy-season precipitation is zero within error. About 10% of rainy-season precipitation accounts for four fifths of growing-season streamflow, and the remaining (46%) rainy-season precipitation accounts for about half of dormant-season streamflow. About three fourths of snowy-season precipitation becomes dormant season streamflow, and perhaps one sixth eventually evapotranspires (but this is zero within error). A small but well-defined proportion is also carried over to the growing season, accounting for one fifth of growing-season streamflow. All quantities are shown ± standard errors. Widths of lines are approximately proportional to water fluxes. Fluxes within one standard error of zero are shown by dashed lines.**





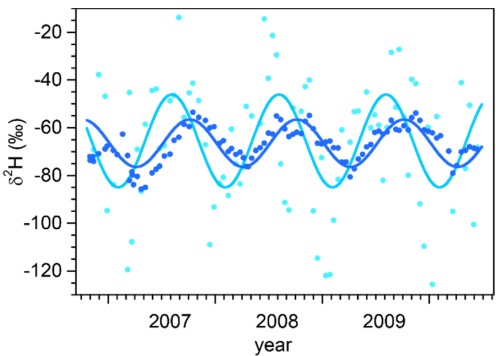

1040

**Figure 9. Deuterium time series in bi-weekly bulk samples of precipitation (light blue) and grab samples of streamwater (dark blue), with superimposed seasonal sinusoidal cycles fitted by volume-weighted least squares. The vertical axis has been expanded to better show the seasonal cycles, with the result that several precipitation values are not shown. The amplitudes of the fitted seasonal cycles are $A_P = 19.4 \pm 3.4‰$ and $A_S = 8.7 \pm 0.9‰$ in precipitation and streamflow, respectively, implying that the** 1045 **flow-weighted young water fraction (the fraction of discharge that is younger than approximately 2-3 months) is $F^*_{yw} = A_S/A_P = 0.45 \pm 0.09$. Rescaling $F^*_{yw}$ by the ratio between the average annual discharge and precipitation fluxes yields the flow-weighted young water fraction of precipitation (the fraction of precipitation that is discharged in less than approximately 2-3 months), $^P F^*_{yw} = F^*_{yw} \overline{Q}/\overline{P} = 0.29 \pm 0.06$.**



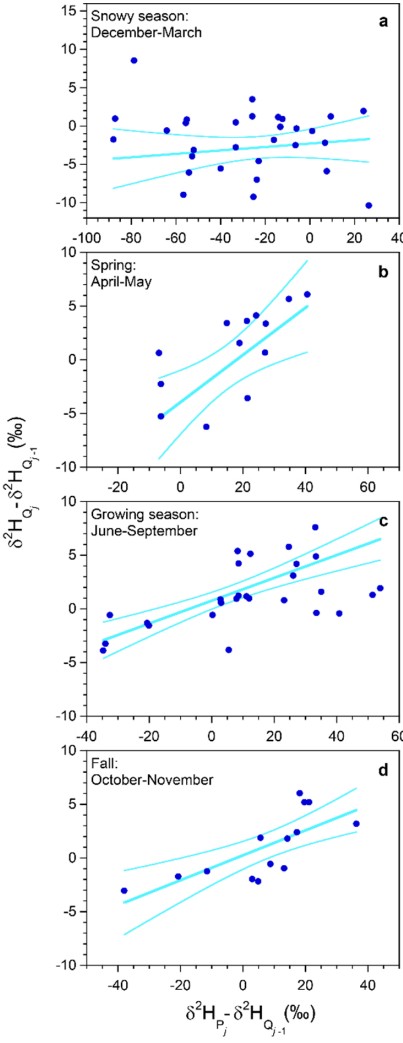

**Figure 10. Ensemble hydrograph separation using biweekly isotope measurements at Hubbard Brook Watershed 3. Straight lines show least-squares regressions weighted by cumulative stream discharge over each two-week sampling interval. Curved lines indicate 95% confidence bounds for the fits. The regression slopes yield ensemble estimates of the biweekly volume-weighted new water fraction of discharge (the volume fraction of discharge that originated from precipitation that fell in the previous two-week sampling interval); $^{Q}F^{*}_{new}$ = 0.022±0.033 during the snowy season (December-March, panel a), 0.220±0.078 during the spring (April and May, panel b), 0.106±0.028 during the growing season (June-September, panel c), and 0.116±0.034 during the fall (October and November, panel d). Rescaling these biweekly event new water fractions by the ratio between seasonal discharge and seasonal precipitation yields the biweekly volume-weighted new water fractions of precipitation (the volume fraction of precipitation that leaves as discharge within the following two-week sampling interval); $^{P}F^{*}_{new}$ = 0.015±0.022 during the snowy season, 0.311±0.111 during the spring, 0.027±0.007 during the growing season, and 0.076±0.023 during the fall. Axes vary from panel to panel but their ratios are held constant, so the plotted lines correctly depict the relative steepness of the regression slopes.**



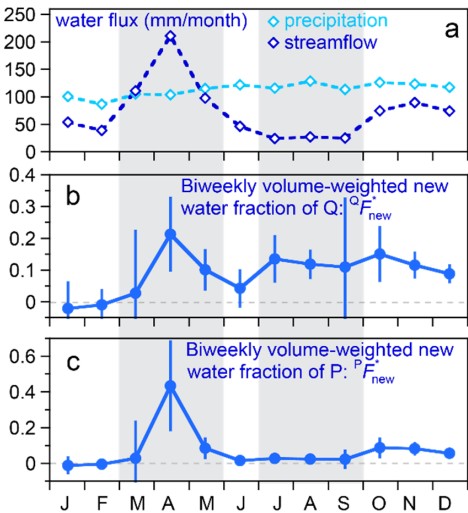

**Figure 11. Seasonal patterns in (a) average precipitation and streamflow fluxes, (b) biweekly volume-weighted new water fractions of streamflow $^{Q}F_{new}^{*}$ (fraction of streamflow derived from precipitation that fell in the previous two weeks), and (c) biweekly volume-weighted new water fractions of precipitation $^{P}F_{new}^{*}$ (fraction of precipitation that becomes streamflow within the following two weeks), as determined from ensemble hydrograph separation (Eqs. 32 and 33; Fig. 10). Dashed lines in (b) and (c) indicate new water fractions of zero. Average precipitation fluxes (a) vary little from month to month, whereas average streamflow fluxes show clear high flows resulting from snowmelt from March through May and clear low flows attributable to evapotranspiration losses from July through September. Both intervals are marked by gray shading. Ensemble hydrograph separations imply that recent (previous two weeks) precipitation comprises about 20% of streamflow during the snowmelt peak in April, roughly 0% (within error) during the cold winter months of January, February, and March, and roughly 10% (within error) during the rest of the year. These streamflow fractions can be re-expressed as fractions of precipitation, by multiplying by monthly streamflow and dividing by monthly precipitation. The resulting biweekly new water fractions of precipitation quantify the fractions of precipitation that leave the catchment as streamflow within the following two weeks (c). These are zero within error in January, February, and March, rise to 43% during April snowmelt, decline to 2% or less throughout the growing season (June through September), and then rise to 5-9% during October, November, and December.**



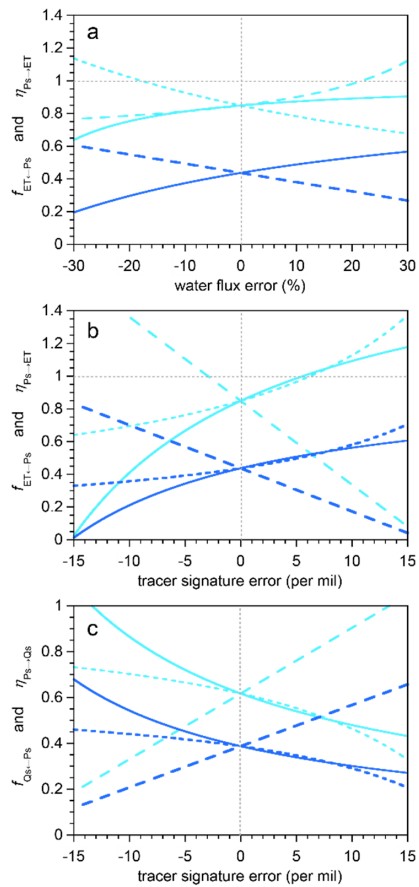

**Figure 12. Sensitivity of end-member splitting fractions to measurement errors in water fluxes (a) and tracer signatures (b,c). Light blue curves show variations in the fraction of evapotranspiration ($f_{ET \leftarrow P_s}$; a,b) and summer streamflow ($f_{Q_s \leftarrow P_s}$; c) that originates as summer precipitation. Dark blue curves show variations in the fraction of summer precipitation that eventually evapotranspires ($\eta_{P_s \rightarrow ET}$; a,b) or becomes summer streamflow ($\eta_{P_s \rightarrow Q_s}$; c). Solid curves show effects of errors in $P_s$ (a) and $\overline{\delta}_{P_s}$ (b,c). Dotted curves show effects of errors in $P_w$ (a) and $\overline{\delta}_{P_w}$ (b,c). Dashed curves show effects of errors in $Q$ (a), $\overline{\delta}_Q$ (b), and $\overline{\delta}_{Q_s}$ (c). Curves are calculated using Eqs. (11), (18), (22), and (25), using input values from Fig. 4, adjusted as shown on the x-axis of each panel.**

1080

1085

1090