# Peer review of "Seasonal partitioning of precipitation between streamflow and evapotranspiration, inferred from end-member splitting analysis"

_Hydrology and Earth System Sciences, 2019_

## Referee Comment (RC1) · Pertti Ala-aho (Referee) · 20 Sep 2019

Review for manuscript Manuscript ID: HESS-2019-420 – Seasonal partitioning of precipitation between streamflow and evapotranspiration, inferred from end-member splitting analysis

Best authors and editors, Thank you for the opportunity to review this interesting manuscript. Authors present a concept of "end-member splitting" to study partitioning of precipitation to different fates (streamflow or evapotranspiration) and how this

partitioning varies in different seasons. To demonstrate the use and usefulness of their concept, they use a long-term hydrometric and water isotope dataset from the well-studied Hubbard Brook experimental watershed. With the analysis, the authors are able to analyse percentages of precipitation (importantly with uncertainty estimates) that end up in different hydrological fluxes in different hydrological seasons, even in different months. They compare the resulting partitioning with young water fractions, new water fractions using the same dataset, and report roughly similar outcomes from different analysis.

In my opinion the work is of great interests to the hydrological community. The work brings forward a data-based technique to study how water is stored in catchments over seasons, which has so far been difficult to demonstrate without numerical modelling. Also, the analysis gives a new handle on seasonal water sources for evapotranspiration, which I think is ecohydrologically highly relevant. Even though the authors modestly don't want to emphasise novelty of the work, I think the analysis and examples they present a great framework (and an example dataset after the data and scripts gets deposited and made available) to run similar analysis in other regions of the world and improve our understanding about catchment-scale hydrological partitioning. As similar water isotope data are increasingly being collected in different environments, the analysis presented here could be readily transferred elsewhere. My only concerns are related to how snow processes are accounted for, see comments for details.

The paper is well written, although the structure of the paper is unorthodox; equations in the introduction, methods and results combined in a long "proof of concept" chapter. However, I think the paper flows well, probably better than cramming all equations to a methods chapter. In my reading the calculations are valid, and figures are of good quality. I recommend the manuscript to be published, and offer some suggestions for further consideration.

Sincerely, Pertti Ala-aho

comments: The only real concern I have is related to winter isotope end-member in you proof-of-concept analysis: you have samples for winter precipitation (snowfall) but the water that catchment receives to further partition between streamflow and ET is snowmelt. There is evidence that snowmelt water is enriched in heavy isotopes compared to cumulative snowfall (Koeniger et al. 2008, Claassen et al. 1995, Ala-aho et al. 2017, Earman et al 2006). Particularly so for catchments with significant snow interception, which I understand is the case in Hubbart Brook catchment (Penn et al 2012). The (few) studies reporting canopy-influenced snow enrichment over the whole snow season show enrichment of 2-3 ‰ in d18O, translating to ∼15-25 ‰ in d2H. I suppose you could test for snowpack isotope enrichment in Hubbard Brook with snowmelt data from Hooper and Shoemaker (1986)? If your winter precipitation end-member would indeed turn out biased because of snowpack enrichment in heavy isotopes, this could make the isotope signature in your winter end-member pretty close to what you have for the dormant season (Fig. 3a and 3b). With regard to your proof-of-concept analysis, I guess such shift would influence your reported partitioning percentages and related uncertainties (more winter signal and more uncertainty throughout the system?). More generally, this would bring uncertainties in site comparisons between snow-influenced and snow-free catchments. You touch upon biased sampling of end-members on P22L654, and I would like to see further discussion if and how systematic bias caused by heavy isotope enrichment in seasonal snowpack might influence your analysis.

P3 L60: if by "this example" you refer to the proof-of-concept analysis in the paper, to be precise you study where winter season precipitation, not snowmelt, ends up.

P4 EQ7: Q_a*n_A->M: M should be X?

P12 L342-P13 L371: not sure how interesting this analysis is, in context of the message you put forward in the paper. I suggest to consider if this analysis is crucial to your work - leaving it out would further streamline the paper.

L373: I would not think it is striking that large part of the snowy season precipitation leaves the catchment during rainy season because of snowmelt in April and May happens in the rainy season, as you discuss later on in the chapter.

P13 L378 do you imply that ~30% of winter precipitation is lost to evaporation/sublimation over the snow season? According Penn et al. (2012) seems like snow interception sublimation can account for 10-30% of snowfall in the region (not accounting for sublimation from ground snow), and numbers alike are typically reported (Varhola et al 2010). Relating this with your ET composed of snowy season precipitation 18 +-18% in Fig. 4, bulk of winter-sourced ET would be from snow sublimation (sublimation by definition sourced from winter precipitation), which leaves little to none winter precipitation for summer ET. This does not seem intuitive, referring to your very nice Allen et al. (2019) paper. From the water balance equation (13) sublimation should be embedded in the total ET, but to me the numbers don't add up. Can you please clarify how your analysis takes snow sublimation into account?

Fig.6: typo in the caption, Figure x6

L548: do you mean "precipitation", instead of "discharge"?

P20L596: "we" instead of "I"?

P26 L770-779: I was surprised by the discrepancy between your analysis the Allen et al (2019) results, even though the environment is different. Looking forward to the end-member splitting analysis with xylem and soil water data.

References: Ala-aho, P., Tetzlaff, D., McNamara, J., Laudon, H., Soulsby, C. ,2017. Modeling the isotopic evolution of snowpack and snowmelt: testing a spatially distributed parsimonious approach. Water Resour. Res., 53, 5813-5830, DOI:10.1002/2017WR020650 Koeniger, P., J. A. Hubbart, T. Link, and J. D. Marshall (2008), Isotopic variation of snow cover and streamflow in response to changes in canopy structure in a snow-dominated mountain catchment, Hydrol. Processes, 22,

557–566, doi:10.1002/hyp.6967. Claassen, H., and J. Downey (1995), A model for deuterium and oxygen 18 isotope changes during evergreen interception of snow-fall, Water Resour. Res., 31, 601–618, doi:10.1029/94WR01995. Earman, S., A. R. Campbell, F. M. Phillips, and B. D. Newman (2006), Isotopic exchange between snow and atmospheric water vapor: Estimation of the snowmelt component of ground-water recharge in the southwestern United States, J. Geophys. Res., 111, D09302, doi:10.1029/2005JD006470. Varhola, A., N. C. Coops, M. Weiler, and R. D. Moore (2010), Forest canopy effects on snow accumulation and ablation: An integrative re-view of empirical results, J. Hydrol., 392, 219–233, doi:10.1016/j.jhydrol.2010.08.009. Penn, C. A., Wemple, B. C. and Campbell, J. L. (2012), Forest influences on snow ac-cumulation and snowmelt at the Hubbard Brook Experimental Forest, New Hampshire, USA. Hydrol. Process., 26: 2524-2534. doi:10.1002/hyp.9450 Allen, S. T., Kirchner, J. W., Braun, S., Siegwolf, R. T. W., and Goldsmith, G. R.: Seasonal origins of water used by trees, Hydrol. Earth Syst. Sci., 23, 1199-1210, doi: 10.5194/hess-23-1199-2019, 2019.

---

## Referee Comment (RC2) · Sylvain Kuppel (Referee) · 24 Sep 2019

In this paper, Kirchner and Allen present and apply a methodology they call "end-member splitting", which focuses (as the name suggests) on tracking the fate of water inputs (here, precipitation) between the different outputs considered (here, streamflow and evapotranspiration). The authors use a publicly available dataset from the Hubbard Brook long-term experimental catchment, which provide the hydrometric and tracer (here $\delta^{18}O$) measurements needed for this method. Distinguishing several sub-annual time periods across the year (e.g. snowy/rainy season, dormant/growing season) with

distinctive isotopic signature in precipitation, they evidence significant inter-seasonal carryover of precipitation (P) inputs into streamflow, implying transient storage at catchment scale. The information available for evapotranspiration (ET) shows by contrast that ET (mostly limited to rainy season) is mostly supplied by P from this same rainy season. This analysis is jointly conducted with the more traditional end-member mixing analysis and other recent metrics such as "young water fraction" and "new water fraction", providing complimentary views on how water transits in the studied catchment. Limits and sources of uncertainty of the method are also discussed, and potential applications outlined.

I really enjoyed reading this manuscript, which is also well-written. By seeking to answer the question "where will this precipitation go?", this approach importantly complements the more traditional end member mixing approach ("what is this streamflow made of?"). Crucially (and as also noted by referee P. Ala-Aho) this forward approach gives tools to track which precipitation is most likely to be evapotranspired, hereby helping predicting ecosystem response to changing precipitation patterns.

The elegance of this data-driven technique obviously comes with limitations. In particular (and this is shared with end member mixing) if some components of the water balance are overlooked (e.g. interannual carryover, significant groundwater inflow/outflow, see Fan, 2019) or if fractionation processes are significant. Such limitations are however often present in the more or less complex modelling approaches currently providing the basis for "forward-tracking" water across the landscape. Finally, although the authors highlight that they "only" combine existing methods, the ready-to-use dataset and promised R routine make it quickly beneficial for widespread test, discussion, and use in the community.

To summarize, I find it to be a very significant contribution to the field and recommend it for publication in HESS. I only have a few very minor/technical comments, easy to address.

**Specific comments**

- **L298-299**: The link with tree rings records is not obvious, maybe add a few words?

- **L378**: Please provides a reference for the SWE estimates

- **L456**: It can actually be a combination of both

- **L541-554**: The added values of the young water fraction comparison is not obvious to me. As the authors write right after, precise mathematical comparison is not possible, given the numerous unknown overlaps between season length and the reference 2-3 months. As a result, I find it hard to see the consistency of the 0.45±0.09 young water fraction with a 55±19% of intra-season contribution over a 4-month season AND a 62±9% of intra-season contribution over a 8-month season... Consider removing this analysis, or restrain it to the shortest seasons. Note that such limitation is much less significant for the new water fraction (as interestingly used in the next section), given the much shorter time step involved (as compared to seasons' lengths).

- **L570-575**: Considering giving more details about $\alpha$ and $\epsilon_j$ in Eq. (38), so that readers can get the "main" picture without necessarily reading Kirchner (2019).

- **L607-609**: I am not sure to understand how one can link "10% of growing season streamflow is less than 2 weeks old" to "half of growing season streamflow comes from P in that same 4-month season".

**Technical comments**

- **Eqs. 26, 27, 28, 29**: I think there is a typo in the second member with $\eta$ instead
of $\Delta\eta$

- **L411**: To help the reader, consider starting this sentence with "Regarding summer precipitation, the monthly end-member [....]"

- **L411**: typo: "streamflow" instead of "rainy-season precipitation"

- **L596**: "We" instead of "I" ?

- **Fig12** This figure has a lot of different color/line type codes, making it complex to read. Consider adding (colored) mention of $f$ or $\eta$ directly on the graphs

**Reference**

Fan, Y.: Are catchments leaky?. Wiley Interdisciplinary Reviews: Water, e1386, 2019.
Kirchner, J. W.: Quantifying new water fractions and transit time distributions using

ensemble hydrograph separation: theory and benchmark tests, Hydrol. Earth Syst. Sci., 23, 303-349, doi: 10.5194/hess-23-303-2019, 2019.

---

## Author Response (AR1)

We thank Dr. Ala-aho for his thoughtful review of our manuscript.

The main issue that Dr. Ala-aho identifies is the possibility that isotopic fractionation during snowpack sublimation could bias our results (or indeed any isotopic results that rely on sampling of snowfall rather than snowmelt). We agree that snowpack fractionation is a potential confounding factor, particularly where a significant fraction of snowfall is intercepted by the canopy and subsequently fractionates. In the revised manuscript we will mention this possibility. In the case of the Hubbard Brook analysis, however, we expect that canopy interception and snowpack fractionation are much smaller than the 15-25 per mil suggested by Dr. Ala-aho, for three reasons.

First, interception and sublimation losses at Wastershed 3 are likely to be small. Penn et al. (2012) found significant interception at Hubbard Brook in conifer forests at higher altitudes on north-facing slopes, but Watershed 3 is a deciduous forest at lower altitudes on a south-facing slope. For the south-facing slopes at Hubbard Brook, which are dominated by deciduous trees, Penn et al. found no statistically significant difference in snowpack accumulation between forest plots and adjacent clearings (mean difference 1.41 cm, standard error 1.4 cm, p=0.3). This is less than 10 percent of peak SWE and only about 3 percent of average winter precipitation.

Second, most of the studies that have been used to quantify canopy interception and sublimation losses, such as those reviewed by Varhola et al. (2010), involved conifer forests and therefore much higher rates of canopy interception than one would expect from the bare branches of a wintertime deciduous forest like that of Watershed 3 at Hubbard Brook. Most of these sites are also in the arid west of North America, where one would expect greater sublimation losses than in the more humid northeast, where Hubbard Brook is situated.

Third, the isotope data plotted in Fig. 2b show no evidence of evaporative fractionation; indeed, if anything the streamwater plots slightly above the local meteoric water line rather than below, as one would expect if there were significant moisture loss due to evaporation or sublimation.

Below we respond (in bold type) to Dr. Ala-aho's specific comments (in normal type).

P3 L60: if by "this example" you refer to the proof-of-concept analysis in the paper, to be precise you study where winter season precipitation, not snowmelt, ends up.

**Correct. We will change "snowmelt" to "precipitation".**

P4 EQ7: Q_a*n_A->M: M should be X?

**Nice catch! We will fix this.**

P12 L342-P13 L371: not sure how interesting this analysis is, in context of the message you put forward in the paper. I suggest to consider if this analysis is crucial to your work - leaving it out would further streamline the paper.

**We did consider this. Although it is not a main point, we do think that in some contexts it may be helpful to compare end-member mixing and end-member splitting results to a "null model" in which no source, or no output, is preferred over another. Thus we have provided the mathematical background for such null model comparisons.**

L373: I would not think it is striking that large part of the snowy season precipitation leaves the catchment during rainy season because of snowmelt in April and May happens in the rainy season, as you discuss later on in the chapter.

We actually discuss this two sentences later.  Of course inter-seasonal transfer from the snowy season to the rainy season via snowpack accumulation and melt is not particularly surprising, but the magnitude is still worth knowing.  And given that 18% (+/-3%) of summer precipitation is carried over to become winter streamflow (and this must be groundwater, since there is no snowpack storage of summer rainfall), it is likely that a substantial amount of the carryover of winter precipitation also takes place via groundwater storage, rather than snowpack.  This illustrates the potential of our approach: without tracer data, and the right tools to interpret them, inter-seasonal carryover in groundwater storage would remain invisible.

P13 L378 do you imply that _30% of winter precipitation is lost to evaporation/sublimation over the snow season? According Penn et al. (2012) seems like snow interception sublimation can account for 10-30% of snowfall in the region (not accounting for sublimation from ground snow), and numbers alike are typically reported (Varhola et al 2010). Relating this with your ET composed of snowy season precipitation 18 +-18% in Fig. 4, bulk of winter-sourced ET would be from snow sublimation (sublimation by definition sourced from winter precipitation), which leaves little to none winter precipitation for summer ET. This does not seem intuitive, referring to your very nice Allen et al. (2019) paper. From the water balance equation (13) sublimation should be embedded in the total ET, but to me the numbers don't add up. Can you please clarify how your analysis takes snow sublimation into account?

Your numbers don't add up because you are assuming that any winter precipitation that does not appear in the snowpack must have fallen as snow and must have been lost to evaporation and sublimation, and those assumptions are not correct at Hubbard Brook.  At Hubbard Brook, there is significant rainfall and snowmelt during winter, so one cannot compare snowpack accumulation to winter precipitation and attribute the difference to evaporation or sublimation.  Trying to estimate evaporation or sublimation by mass balance is made even more difficult by inter-seasonal groundwater storage.  That is why end-member splitting only uses long-term mass balances, and only infers seasonal contributions to total annual ET (not ET losses in individual seasons).  The end-member splitting result is 18±18 percent for total ET losses, at any time of the year, that originate as winter precipitation.

Penn et al.'s data for south-facing, deciduous forests at Hubbard Brook (which would apply to Watershed 3) show only 1.4±1.4 cm of evaporation or sublimation from the snowpack (Penn et al., 2012, p. 2530).  This is statistically indistinguishable from zero, and is only 7% of the average SWE (19 cm) in south-facing forest plots on Penn et al.'s March 4[th] sampling date.  Furthermore, 1.4 cm of evaporation or sublimation would be only 3% of the long-term average annual precipitation that falls during the snowy period from December through March (40 cm), and only about 1% of total annual precipitation.  Of the 33 studies summarized by Varhola et al. (2010), all but two examined snow accumulation in conifer forests.  Thus Varhola et al.'s results greatly exaggerate the degree of canopy interception and sublimation that should be expected in deciduous forests like Watershed 3 at Hubbard Brook.

Fig.6: typo in the caption, Figure x6

Sorry!  We'll fix it.

L548: do you mean "precipitation", instead of "discharge"?

Yes we do, thanks!  We'll fix that.

P20L596: "we" instead of "I"?

Another nice catch.  We'll fix it.

P26 L770-779: I was surprised by the discrepancy between your analysis the Allen et al (2019) results, even though the environment is different. Looking forward to the end-member splitting analysis with xylem and soil water data.

**We are intrigued by this too.  We do not necessarily expect similar results from Hubbard Brook and the Swiss sites, since (as we wrote) the potential for soil water storage is quite different in the two settings.**

We thank Dr. Kuppel for his thoughtful review of our manuscript. Below we respond (in bold type) to Dr. Kuppel's specific comments (in normal type).

• L298-299: The link with tree rings records is not obvious, maybe add a few words?

**We will clarify this. The point here is that tree-ring isotope values are often assumed to reflect the isotopic composition of either growing-season precipitation or annual average precipitation, but the seasonal sources of xylem water (and thus of tree-ring isotopes) may vary with climate and subsurface moisture storage characteristics. Thus, being able to infer the isotopic composition of the transpiration flux would provide an additional constraint for calibrating tree-ring isotopes.**

• L378: Please provides a reference for the SWE estimates

**Will do (it's Campbell et al., 2010, cited earlier in the paper).**

• L456: It can actually be a combination of both

**Correct. When we used "or", we meant it in the mathematical sense (which includes "or both"), but we can be more explicit about this.**

• L541-554: The added values of the young water fraction comparison is not obvious to me. As the authors write right after, precise mathematical comparison is not possible, given the numerous unknown overlaps between season length and the reference 2-3 months. As a result, I find it hard to see the consistency of the 0.45±0.09 young water fraction with a 55±19% of intra-season contribution over a 4-month season AND a 62±9% of intra-season contribution over a 8-month season... Consider removing this analysis, or restrain it to the shortest seasons. Note that such limitation is much less significant for the new water fraction (as interestingly used in the next section), given the much shorter time step involved (as compared to seasons' lengths).

**The point here is that we have two different lines of analysis that both show significant (40-55 percent) fractions of streamflow originate as precipitation within the previous 2-4 months, and that, unsurprisingly, this fraction becomes bigger as one considers longer and longer intervals of time. (One could view this as an approximation of the cumulative transit time distribution, for example). The point is that it did not have to turn out this way. If one or the other method was sufficiently ill-conceived, one could have gotten results that made no sense together. And considered together, all of these lines of evidence point toward significant inter-seasonal carryover of water in catchment storage (as we say in the last paragraph of Section 2.7). We can add a couple of sentences of further explanation here.**

• L570-575: Considering giving more details about α and ε_j in Eq. (38), so that readers can get the "main" picture without necessarily reading Kirchner (2019).

**Alpha is the intercept and epsilon_j is the error term in this linear regression equation. We thought that most readers would understand these terms since Eq. (38) has already been identified as a linear regression equation, but we can certainly define them here too.**

• L607-609: I am not sure to understand how one can link "10% of growing season streamflow is less than 2 weeks old" to "half of growing season streamflow comes from P in that same 4-month season".

**One way is by considering the counterfactual case: what if our results showed that, say, 80% of growing-season streamflow was less than two weeks old (from ensemble hydrograph separation) and that half of growing-season streamflow came from precipitation in the same 4-month season?  Then clearly something would be wrong with one or both methods, since the amount of water that is less than four months old must exceed the amount of water that is less than two weeks old (since all of the latter is also part of the former).  Instead, our results show that during the growing season the biweekly new water fraction (7.8-13.4 percent, within one standard error) is small enough to be broadly consistent with the end-member mixing estimate that roughly half (36-74 percent, within one standard error) of growing-season streamflow originates from growing-season precipitation.  We will slightly expand on our discussion of these points in the text.**

Technical comments

• Eqs. 26, 27, 28, 29: I think there is a typo in the second member with η instead of Δη

**Yikes!  You are right.  Such are the perils of copy-paste errors during late-night equation editing.  Thanks for catching those.  We'll fix them (and check all the equations once again).**

• L411: To help the reader, consider starting this sentence with "Regarding summer precipitation, the monthly end-member [....]"

**Instead, we will just include the corresponding expressions for both summer and winter precipitation end-members.**

• L411: typo: "streamflow" instead of "rainy-season precipitation"

**This typo isn't on L411, but instead on L433.  We'll fix it.**

• L596: "We" instead of "I" ?

**Dr. Ala-aho caught this one too.  We'll fix it.**

• Fig12 This figure has a lot of different color/line type codes, making it complex to read. Consider adding (colored) mention of f or η directly on the graphs

**We understand that this figure is complex and takes some effort to decode.  We tried to simplify it in several different ways, and they only made things worse.  We can of course color-code the f and η symbols on the y-axes.  However, it isn't possible to annotate each curve on each diagram; we've tried, and there just isn't space.  The interpretation would be simpler if we made each panel into a separate figure, or panel (a) into one figure and (b) and (c) into a second figure.  But we think that this sensitivity analysis isn't important enough to justify two or three figures.**

[revised manuscript text omitted]
_\mathrm{s} = q_{\mathrm{P_s} \to \mathrm{Q_s}} + q_{\mathrm{P_w} \to \mathrm{Q_s}} \quad , \qquad Q_\mathrm{w} = q_{\mathrm{P_s} \to \mathrm{Q_w}} + q_{\mathrm{P_w} \to \mathrm{Q_w}} \tag{9}$$

where $Q_\mathrm{s}$ and $Q_\mathrm{w}$ represent the average annual sums of stream discharge during the summer and winter seasons, and (for example) $q_{\mathrm{P_s} \to \mathrm{Q_s}}$ and $q_{\mathrm{P_w} \to \mathrm{Q_s}}$ are the average annual fluxes of summer streamflow that originated as summer and winter precipitation, respectively. Equation (9) directly implies that, no matter how the precipitation end-members are defined, they must jointly account for all the precipitation that could eventually become streamflow (including, potentially, precipitation in multiple previous summers or winters). In other words, streamflow must be composed only of a mixture of the summer and winter precipitation, $P_\mathrm{s}$ and $P_\mathrm{w}$; there can be no other end members, sampled or not (although obviously streamflow can contain flows from various catchment compartments in which summer and winter precipitation have been stored and mixed). We also assume isotopic mass balance for the water that eventually becomes discharge,

$$Q_\mathrm{s}\, \bar{\delta}_{\mathrm{Q_s}} = q_{\mathrm{P_s} \to \mathrm{Q_s}}\, \bar{\delta}_{\mathrm{P_s}} + q_{\mathrm{P_w} \to \mathrm{Q_s}}\, \bar{\delta}_{\mathrm{P_w}} \quad \text{and} \qquad Q_\mathrm{w}\, \bar{\delta}_{\mathrm{Q_w}} = q_{\mathrm{P_s} \to \mathrm{Q_w}}\, \bar{\delta}_{\mathrm{P_s}} + q_{\mathrm{P_w} \to \mathrm{Q_w}}\, \bar{\delta}_{\mathrm{P_w}} \quad , \tag{10}$$

where $\bar{\delta}_{\mathrm{Q_s}}$, $\bar{\delta}_{\mathrm{Q_w}}$, $\bar{\delta}_{\mathrm{P_s}}$, and $\bar{\delta}_{\mathrm{P_w}}$ are the volume-weighted average isotopic signatures in summer and winter streamflow and precipitation. Equation (10) implies that the precipitation that eventually becomes streamflow does not undergo substantial isotopic fractionation (the effects of which are discussed further in Sect. 3.3). It does not imply that no such fractionation occurs in the water fluxes that are eventually evapotranspired (and in any case, evapotranspiration fluxes are neither sampled nor directly measured). Combining Eqs. (9) and (10) yields the end-member mixing equations for summer streamflow,

$$f_{\mathrm{Q_s} \leftarrow \mathrm{P_s}} = \frac{q_{\mathrm{P_s} \to \mathrm{Q_s}}}{Q_\mathrm{s}} = \frac{\bar{\delta}_{\mathrm{Q_s}} - \bar{\delta}_{\mathrm{P_w}}}{\bar{\delta}_{\mathrm{P_s}} - \bar{\delta}_{\mathrm{P_w}}} \qquad \text{and} \qquad f_{\mathrm{Q_s} \leftarrow \mathrm{P_w}} = \frac{q_{\mathrm{P_w} \to \mathrm{Q_s}}}{Q_\mathrm{s}} = \frac{\bar{\delta}_{\mathrm{Q_s}} - \bar{\delta}_{\mathrm{P_s}}}{\bar{\delta}_{\mathrm{P_w}} - \bar{\delta}_{\mathrm{P_s}}} \quad , \tag{11}$$

where $f_{\mathrm{Q_s} \leftarrow \mathrm{P_s}}$ and $f_{\mathrm{Q_s} \leftarrow \mathrm{
[revised manuscript text omitted]

where $P_\mathrm{s}$ and $P_\mathrm{w}$ represent the average annual sums of precipitation falling in the summer and winter, respectively, $Q$ represents annual average discharge, and $ET$ represents average annual evapotranspiration. Equation (13) assumes that these fluxes are much larger than any other inputs (such as direct surface condensation or groundwater inflows) or outputs (such as groundwater outflow). Equation (13) is also assumed to hold over time scales long enough that changes in catchment
storage are trivial compared to the cumulative input and output fluxes. These same assumptions are invoked in hydrometric studies that infer $ET$ from long-term catchment water balances (e.g., Vadeboncoeur et al., 2018). However, such hydrometric studies cannot reliably estimate the seasonal origins of evapotranspiration, because changes in catchment storage may be substantial on seasonal time scales.

We can straightforwardly apply end-member mixing to the total annual discharge, analogously to the approach used in Eqs. (9)-(11) for discharge during the individual seasons. All discharge must originate as either summer or winter precipitation, and thus

$$Q = q_{P_\mathrm{s} \to Q} + q_{P_\mathrm{w} \to Q} \qquad , \qquad (14)$$

where $q_{P_\mathrm{s} \to Q}$ and $q_{P_\mathrm{w} \to Q}$ are the annual average fluxes that originate as summer and winter precipitation. Isotopic mass
balance for the water that eventually becomes discharge implies

$$Q\,\bar{\delta}_Q = q_{P_\mathrm{s} \to Q}\,\bar{\delta}_{P_\mathrm{s}} + q_{P_\mathrm{w} \to Q}\,\bar{\delta}_{P_\mathrm{w}} \qquad , \
[revised manuscript text omitted]